# Active Faults Sources for the Pátzcuaro-Acambay Fault System (Mexico): Fractal Analysis of Slip Rates, and Magnitudes $M_w$ Estimated from Fault Length.

Avith Mendoza-Ponce[1,4], Angel Figueroa-Soto[2,4], Diana Soria-Caballero[1,4], and Víctor Hugo Garduño-Monroy[3,4]

[1]Posgrado en Ciencias de la Tierra, Escuela Nacional de Estudios Superiores, UNAM, Antigua Carretera a Pátzcuaro No.8701, Morelia, Michoacán, Mexico
[2]CONACyT - Instituto de Investigaciones en Ciencias de la Tierra, Universidad Michoacana de San Nicolás Hidalgo
[3]Instituto de Investigaciones en Ciencias de la Tierra, Universidad Michoacana de San Nicolás Hidalgo, Morelia, Michoacán, Mexico
[4]Centro Mexicano de Innovación en Energía Geotérmica (CeMIE-Geo)-Proyecto 17, Carretera Tijuana-Ensenada No. 3918, Zona Playitas, 22860, Ensenada, B.C., Mexico

*Correspondence to:* Angel Figueroa-Soto (angfsoto@gmail.com)

**Abstract.** The Pátzcuaro-Acambay Fault System (PAFS), located in the central part of the Trans-Mexican Volcanic Belt (TMVB), is delimited by an active transtensive deformation area associated with the oblique subduction zone between the Cocos and North American plates, with a convergence speed of 55 mm/yr at the latitude of the state of Michoacán, Mexico. Part of the oblique convergence is transferred to this fault system, where the slip rates range from 0.009 to 2.78 mm yr$^{-1}$. This

has caused historic earthquakes in Central Mexico, such as the Acambay quake ($M_s = 6.9$) on November 19, 1912 with surface rupture, and another in Maravatío in 1979 with $M_s = 5.6$. Also, paleoseismic analyses are showing Quaternary movements in some faults, with moderate to large magnitudes. Notably, this zone is seismically active, but lacks a dense local seismic network, and more importantly, its neotectonic movements have received very little attention. The present research encompasses three investigations carried out in the PAFS. First, the estimation of the maximum possible earthquake magnitudes, based on

316 fault lengths mapped on a 15 m Digital Elevation Model, by means of three empirical relationships. In addition, the Hurst exponent $H_w$ and its persistence, estimated for magnitudes $M_w$ (spatial domain) and for 22 slip-rate data (time-domain) by the *wavelet variance analysis*. Finally, the validity of the intrinsic definition of active fault proposed here. The average results for the estimation of the maximum and minimum magnitudes expected for this fault population are $5.5 \leq M_w \leq 7$. Also, supported by the results of $H$ at: the **spatial domain**, this paper strongly suggests that the PAFS is classified in three different

zones (western PAFS, central PAFS and eastern PAFS) in terms of their roughness ($H_w = 0.7, H_w = 0.5, H_w = 0.8$ respectively), showing different dynamics in seismotectonic activity and; the **time-domain**, with a strong persistence $H_w = 0.949$, suggests that the periodicities of slip rates are close in time (process with memory). The fractal capacity dimension ($D_b$) is also estimated for the slip-rate series using the *box-counting method*. Inverse correlation between $D_b$ and low slip-rate concentration was observed. The resulting $D_b = 1.86$ is related to a lesser concentration of low slip-rates in the PAFS, suggesting that larger

faults accommodate the strain more efficiently (length $\geq 3$ km). Thus, in terms of fractal analysis, we can conclude that these 316 faults are seismically active, because they fulfill the intrinsic definition of active faults for the PAFS.

## 1 Introduction

The state of Michoacán in Mexico is an area of high seismic activity, not only due to subduction events, such as the devastating earthquake of 19 September 1985 earthquake ($M_w = 8.1$), but also because of the existence of crustal faults in the interior. Historically, several earthquakes have affected populations such as Pátzcuaro and Araró (in 1845 and 1858), Zinapécuaro and Tlalpujahua (in the 20th century), Acambay (in 1912) and Maravatío (in 1979). More recently, in 2007, a set of earthquakes ($2.5 < M_w < 3.0$) occurred in the vicinity of the city of Morelia, as a consequence of the movement of the active fault named

La Paloma. The major problem here, in central Mexico, is that we are incapable of using seismic and geodesic data of coseismic slip during earthquakes, because we lack a dense local seismic and geodesic network. Indeed, along the PAFS there are only two broad band stations of the Mexican Seismological Service (SSN) in the cities of Morelia (Lat:19.646812, Long:-101.227135) and Acambay (Lat:19.9845, Long:-99.8823). Moreover, the existing paleoseismological studies are too scarce in relation to the number of existing faults.

Above all, this reveals the need to define the intracontinental structures that are susceptible of generating moderate and strong seismic events, and delimit the damaged area that can produce such events, especially in the center of Mexico, which presents highly populated zones. Of course we used the excellent manifestation and geomorphology of faults, and we analyzed the magnitudes $M_w$ derived from fault dimensions and the slip-rate estimations of earlier studies, as well as spatial distribution by Fractal Analysis. In principle, this branch of mathematics gives us a way of describing, measuring and predicting seismic

activity by means of the Hurst exponent and the fractal dimension. We used two main databases: (**a**) 316 average magnitudes $M_w$ calculated from the surface rupture length on a 15 m Digital Elevation Model, and (**b**) 22 slip rates recorded in the literature.

Thus, the goals of this investigation are: (**1**) the estimation of the maximum possible earthquake magnitudes by three empirical relations; (**2**) the definition of a micro-regionalization of the PAFS using the Hurst exponent based on $M_w$ magnitudes;

and (**3**) the validation of our proposed intrinsic definition of Active Fault sources for the PAFS by fractal analysis and semivariograms. Consequently, we are proposing the investigation of the dynamics of the Pátzcuaro-Acambay area, in order to improve territory planning and reduce seismic hazard.

## 2  Tectonic Setting of the PAFS

The Trans-Mexican Volcanic Belt (TMVB) is an active continental volcanic arc that spans cross Mexico with an approximate E-W orientation. The TMVB developed within an extensional tectonics setting resulting from the subduction of the Rivera and Cocos plates beneath the North American plate. The central TMVB is characterized by the Tula-Chapala Fault Zone (Johnson and Harrison, 1990), where the kinematics is extensional and transtensional from the Miocene (Johnson and Harrison, 1989; Martínez-Reyes and Nieto-Samaniego, 1990; Garduño-Monroy et al., 2009) with a left strike slip component (Suter et al., 1992, 1995, 2001; Ego and Ansan 2002; Norini et al., 2006).

Specifically, we will focus on the central and eastern parts of the Tula-Chapala Fault Zone, i.e., the PAFS (Fig. 1, 2). The PAFS is defined as a population of several hundreds of normal faults, oriented E-W and NE-SW, comprising the cities between Pátzcuaro and Acambay(102°-99° W). Its kinematics is summarized as a left-lateral transtensional system with $\sigma3$ trending NW-SE and $\sigma2$ trending NE–SW (Suter et al., 1992, 1995, 2001; Ego and Ansan, 2002; Mennella, 2011). Moreover, according to Mennella (2011) there are three major fault sets in the PAFS, the first and oldest being the NNW-SSE system, expressed mainly by the Tzitzio-Valle de Santiago fault. The other two systems configured lake areas and have the morphology of seismically active faults with E-W and ENE-WSW strikes. Their kinematics show clear evidence from the Miocene (17Ma) with left-strike slip faulting, that later became to normal with a left-lateral component (Suter et al., 2001; Ego and Ansan, 2002; Mennella, 2011). In the NNW-SSE faults, this stress field generates a reactivation as oblique faults with normal right-lateral component. This deformation always keeps the $\sigma3$ moving from 360° N to 340° N.

The Pátzcuaro-Acambay Fault System can be divided into three zones with different geological and geophysical settings: **(1)** The **western PAFS**, between Pátzcuaro and the Tzitzio-Valle de Santiago fault, is an area where three different scenarios have coexisted. First, the andesitic basements of the Miocene (>19 Ma) were in contact with a volcanic sequence characterized by alternation of andesites and ignimbrites, varying in age from 19 to 7 Ma. These volcanic sequences were contemporary with a sinistral strike-slip faulting with E-W and NE-SW structures, which later moved like normal faults (from 12 to 7 My), to finally turn into normal faults with a strike slip component (see Focal Mechanisms, Fig. 2). The complete western zone has a geometry of listric faults with lengths from 3 to 33 km, generating rotations of the Miocene lithological units that allow the rise of hydrothermal fluids. Since the Miocene, this faulting has caused grabens and semi-grabens, causing the formation of lakes. These lake depressions are controlled by old NNW-SSE faults, which act as relay zones today. So, the coexistence of these faults, lacustrine depressions and hydrothermal manifestations make up the second scenario. The last scenario is where monogenetic volcanism is controlled by existing faults; indeed, this volcanism is abundant, and presents NE-SW alignments (Michoacán-Guanajuato Volcanic Field). **(2)** The **central PAFS** extends between the NNW-SSE-trending Tzitzio-Valle de Santiago fault and Maravatío. This sector is basically occupied by the Los Azufres Geothermal Field, which is defined as a volcanic complex with andesitic volcanoes, rhyolitic and dacitic domes, and an important thickness of pyroclastic flows and monogenetic volcanism. In the past million years, magmatic processes have developed, affected by E-W faulting, which also controls the hydrothermal manifestations. Petrological studies show a magmatic chamber located between 4.3 and 9.5 km of depth at the El Guangoche dome (Rangel et al., 2018), probably modifying the fragile ductile limit of the crust. Surely this

modification is responsible for shorter fault lengths ranging from 3 to 26 km. Finally, **(3)** the **eastern PAFS** is mainly formed by the Acambay graben. Its limit with the central zone is defined by the Maravatío area, where the graben is wider (18 km) and the foot wall in the southern sector is formed by Jurassic basement rocks. The hanging wall displays monogenetic volcanism aligned in preferential NW-SE and E-W directions, parallel to the fault where the 1979 earthquake generated ($5.6M_s$). On the other hand, the eastern limit is narrower (14 km), and occupied by complex volcanoes such as the San Pedro volcano, and by small monogenetic volcanoes, all affected by the E-W fault system that generated the 1912 Acambay earthquake ($6.9M_s$). This magnitude is in accordance with fault lengths and with the paleoseismic study of Lacan et al. (2018), in which the longest fault found (47 km) is defined as capable of generating large seismic events ($6.9M_w$). These faults have translational movements and do not generate tilts in the Miocene sequences, as is the case in the Cuitzeo area, therefore, they do not comprise a geothermal flow.

## 2.1 Paleoseismicity in the PAFS

Although the seismotectonic context of the PAFS is summarized as an active left-lateral strike-slip system, seismic hazard studies are still considered incomplete. The reason for this is the scarcity of data regarding the slip rates and recurrence periods of the prehistoric and historical activity of the fault segments within the system. The faults studied with a paleoseismological approach at the eastern portion of the PAFS are: (1) The Acambay-Tixmadejé fault (Urbina and Camacho, 1913; Suter et al., 1995; 1996), where the record of at least five rupture events allowed to reckon a recurrence interval of 3 600 yr, meaning slip rates of 0.17 mm yr$^{-1}$ and potential magnitudes between 6.8$\geq M_w \geq$7. (2) The Pastores fault, with a recurrence interval of 10 000-15 000 yr and 1 100-2 600 yr (short and long time span), with slip rates of 0.03 to 0.23-0.37 mm yr$^{-1}$ and potential magnitudes from 6.6 to 6.8$M_w$ (Suter et al., 1992; Langridge et al., 2013; Ortuño et al., 2015). (3) For the San Mateo fault, Sunye-Puchol et al. (2015) found a recurrence interval of 11 570 yr, a slip rate of 0.085 mm yr$^{-1}$ and potential magnitudes between 6.43 $\geq M_w \geq$ 6.76. (4) The Venta de Bravo fault is capable of producing earthquakes with magnitudes of $M_w \geq$6.9, with a slip rate of 0.22-0.24 mm yr$^{-1}$ and a recurrence interval between 1 940 and 2 390 yr (Lacan et al., 2018). Finally, (5) for the Temascalcingo fault, a current study by Velázquez-Bucio (2018) reports a slip rate of 0.017 mm yr$^{-1}$ and a recurrence of 28 901 yr with a paleo-magnitude of 6.5$M_w$. In addition, Ortuño et al., 2018 reports slip rates ranging from 0.06 $\pm$ 0.02 (minimum long term) to 0.12 $\pm$ 0.02 mm yr$^{-1}$ (maximum value of average short-term).

Other studies using soft-sediment deformations related to seismic activity (seismites) have also been carried out in the basins of the State of Mexico, such as Tierras Blancas (Rodríguez-Pascua et al., 2010) and Ixtlahuaca controlled by the Perales fault (Benente, 2005; Velázquez-Bucio et al., 2013; 2015), which allowed to estimate the potentiality of the nearby faults, obtaining magnitudes of $\geq 6M_w$.

In the state of Michoacán, paleoseismology studies have been concentrated on Pátzcuaro, Morelia and Cuitzeo, where almost a dozen faults were studied in detail (Garduño-Monroy et al., 2001; 2009; Suter, 2016). For these structures, slip-rates were obtained in a range of 0.009-2.78 mm yr$^{-1}$ (long and short time span, respectively), recurrence intervals of 1 200-100 000 yr (long and short time span, respectively) and magnitudes between 5.8 $\geq M_w \geq$ 7.1. Moreover, at the northwest of Morelia,

the structure named Teremendo fault is studied by Soria-Caballero (submitted). Paleoseismic data show slip-rates of 0.11 mm yr$^{-1}$, a time recurrence of 7 726 yr, and potential magnitudes of $5.9 \geq M_w \geq 6.8$.

## 2.2 Historical and Instrumental Seismicity in the PAFS

The Acambay earthquake ($M_s = 6.9$) occurred on November 19, 1912, in the eastern PAFS (Urbina and Camacho, 1913; Suter et al., 1996). The quake killed more than 150 people and caused the destruction of entire villages. During this event, at least three faults showed surface rupture (Urbina and Camacho, 1913): the Acambay-Tixmadejé fault ($D_{max} = 50$ cm), the Temascalcingo fault ($D_{max} = 30$ cm) and the Pastores fault ($29 \leq D_{max} \leq 37$ cm; Ortuño et al., 2015). Subsequently, in 1979, another earthquake with $5.6 M_s$ magnitude and 8.2 km depth (Astiz-Delgado, 1980), caused major damage in Maravatío. In the western zone of the PAFS, some earthquakes have affected populations such as Pátzcuaro and Araró (in 1845 and 1858), and Zinapécuaro and Tlalpujahua (also in the 19th century).

Currently, the eastern PAFS is active with microseismicity, which is documented in the literature (Ego and Ansan, 2002; Campos et al., 2015; Ortuño et al., 2015), and is characterized by a left-lateral transtensive deformation with NW-SE to NNW-SSE orientation. Regarding the west of the PAFS, very close to the city of Morelia, a sequence of seven earthquakes occurred ($2.5 < M_w < 3.0$) with focal mechanisms corresponding to normal faulting with left-lateral components. This set of tremors took place in a 33 hr interval in October 2007, and were recorded by two local stations located within the city (Sing et al., 2012). It is very likely that this sequence of earthquakes was related to the La Paloma fault, considered active, because it affects Holocene deposits (Garduño-Monroy et al., 2009 and Suter, 2016). The rupture of a small segment of this fault can generate earthquakes with magnitudes up to $5 M_w$ (Sing et al., 2012).

The seismic catalog, covering from 1912 to 2018, was obtained from the Seismological Service of Mexico (Servicio Sismológico Nacional, SSN; Fig. 2). The data is available on their web page: www.ssn.unam.mx. The focal mechanism parameters were reported previously by Astiz-Delgado (1980), Suter et al. (1992; 1995), Langridge et al. (2000), Singh et al. (2011; 2012), Rodríguez-Pascua et al. (2012).

## 2.3 GPS Measurements

The multi-temporal comparative study (1998-2003 to 2011) of the dynamics in the eastern zone of the PAFS is presented only by Espinosa-Rodríguez et al., 2016. The vertical tectonic movements show rates ranging from +7.3 to +12.8 mm yr$^{-1}$ in the northern horst of Santa María Tixmadejé, while in the central graben of Acambay they are very weak, of +0.4 to +0.5 mm yr$^{-1}$.

## 3 Materials

### 3.1 Mapping the Pátzcuaro-Acambay Fault System

A fault database was constructed on a 15 m DEM. We used the imagery provided by the Instituto Nacional de Estadística y Geografía (INEGI, acronym in Spanish). We identified and defined fault segments on a Geographic Information System (GIS) on the basis of the excellent morphological evidences. The criterion for the tracing of fault segments was the union of small traces to form a larger one, but only if the geomorphological continuity was clear. The lengths of fault trajectories, which is the main object of study, corresponded to the lengths of mountain front sinuosity, and the scarp was measured at the maximum hillslope value for each fault. We also used the length information of the faults digitalized around the Cuitzeo basin by project 17 of the Centro Mexicano de Innovación en Energía Geotérmica (CeMIEGeo, acronym in Spanish), based in Morelia, Mexico. The assumed error for the morphometric parameters measured here was not relevant for our analysis because the lowest fault length (3000 m) is lesser than map resolution (15 m). Additionally, we are suggesting fault names based on the names of the nearest towns, in order to homogenize nomenclature for researchers interested in correcting or completing the existing database.

Our database consists of 316 faults of $\geq 3$ km length (Fig. 2) and comprises the following characteristics: Fault name; Fault length (meters); Fault scarp height (meters); Begin UTM coordinates (X1, Y1); End UTM coordinates (X2, Y2); Intermediate point UTM coordinates (Xm, Ym) for each trajectory; Close locality name; Distance between locality and fault zone (meters); $M_w$(Wells and Coppersmith, 1994); $M_w$ (Anderson et al., 1996); $M_w$ (Wesnousky. 2008).

### 3.2 Estimation of the Maximum Magnitudes

Maximum and minimum earthquake magnitudes were calculated for the same fault section with three magnitude-scaling relationships. We assessed fault relationships by the surface rupture length (SRL) using Wells and Coppersmith's empirical regression model (1994) for normal faults ($M_w = 4.86 + 1.32 \log_{10}(SRL)$); we also used the equivalent regression model proposed by Wesnousky (2008; $M_w = 6.12 + 0.47 \log_{10}(SRL)$); finally, we included the model proposed by Anderson et al. (1996; $M_w = 5.12 + 1.16 \log(SRL) - 0.2 \log(S)$), where $S$ is the slip rate.

### 3.3 Slip Rates and their Cumulative Distribution

Before applying paleoseismology in Mexico, the slip rates were calculated with the accumulated displacements in the escarpment of each segment and the age of the displaced lithological units. In this sense, we are considering displacement rates of 2 mm yr$^{-1}$, 0.05 mm yr$^{-1}$ and 0.16 mm yr$^{-1}$ for some faults in the PAFS, such as Venta de Bravo, C. El Aguila lava and the Cuitzeo faults, respectively (Suter et al., 1992; 2001). Currently, the paleoseismic analyses of these and other faults of the PAFS have allowed to refine the slip-rate estimates made by Langridge et al. (2000), Garduño-Monroy et al. (2009), Sunye-Puchol et al. (2015), Ortuño et al. (2015), Lacan et al. (2018), Ortuño et al. (2018), Velázquez-Bucio (2018) and Soria-Caballero (submitted). Finally, we were able to analyze 22 slip-rate data derived from these earlier studies (Table 1).

In order to characterize the persistence of the slip-rate series, we constructed a cumulative slip-rate plot, organized from east to west, since most of the microseismicity is concentrated in the eastern PAFS, near the Acambay graben (Rodríguez-Pérez and Zúñiga, 2017). This cumulative plot obeys the well-known Devil's staircase fractal (Fig. 4a). The Devil's staircase is a non-constant but continuously increasing function. It is defined mathematically as the integral of a Cantor set, whose iterative construction implies that the Devil's staircase is a self-similar object. Thus, the fault movements are the physical manifestation of a fractal behavior.

## 4 Methods for the Study of Faults using Fractal Analysis

### 4.1 Self-Similar Behavior in Earth Science

In several works, the geometrical description of patterns of earthquakes, fractures and volcanoes is studied using the self-similar property of fractals. This self-similarity is fulfilled when the objects look geometrically equal on any scale, and is characterized by inverse power laws (Ishimoto and Ida, 1939; Gutenberg and Richter, 1944, 1954; Mandelbrot, 1983; Bak and Tang, 1989; Korvin 1992; Turcotte, 1992; Ghosh and Daemen, 1993; Mazzarini et al., 2010; Pérez-López et al., 2011), where the exponent corresponds to the value of the fractal dimension (i.e. Bak et al., 1987; Tang and Marangoni, 2006). Fractals are irregular, rough, and fragmented objects which display self-similarity (roughness is invariant when scaling). A typical example of fractals is the coastline (coastline paradox). The standard technique to scrutinize scale invariance and estimate fractal dimension is the box counting method.

Self-similarity was studied by Nieto-Samaniego et al. (2005) in the Los Planes fault, Baja California Sur, Mexico, using a detailed fractal analysis of fracture arrays. Their sampled fracture traces have box dimensions between 1.51 and 1.87. Moreover, they proved, for a map of any size, that box dimension is in inverse relation with fracture concentration and in direct relation with fracture density (Renshaw, 1997). They have estimated the fracture concentration as the mean distance between centers of fractures divided by the average fracture lengths (Smirnov and Zavyalov, 1997), which characterizes the interactions between adjacent fractures. Smirnov and Zavyalov (1997) evaluated the critical value of the concentration of ruptures from the standpoint of physics. The failure concentration criterion is a measure of the loss of stability in a set of cracks under stress. If the cracks lose stability, they grow, and coalesce to form larger fractures. As a result, they are sufficiently close to one another, and consequently, a high concentration of cracks appears in certain volumes. Other studies have also shown that the total fracture length is directly proportional to the amount of deformation, i.e., large fractures can accommodate more deformation than small ones (Poulimenos, 2000; Cowie et al., 1995; Ackermann et al., 2001).

### 4.2 The Hurst Exponent

The predictability of time series began with the original work of Harold Edwin Hurst (1951). He focused on the analysis of fluctuating fluvial time series by analyzing the standard deviation of accumulated water flow. Thus, Hurst established the Nile river's rain and drought cycles. These statistics handle the progression of observations in time. The correlation of the past

and the future in the observational time series can be described by the Hurst exponent, $0 \leq H \leq 1$. For independent random processes, with no correlations among samples, $H = 0.5$. For $H > 0.5$, the observational time series is persistence, which means, in average, that the increasing (decreasing) trend in the past induces the continued increasing (decreasing) trend in the future. Persistent time series have a long memory, and a long-term correlation exists between current and future events. On the other hand, when $H < 0.5$, the sequence is characterized by antipersistent behavior. This means that an increasing (decreasing) trend in the past causes a decreasing (increasing) trend in the future. It can be expressed as a time series regression (Xu and Burton, 2006). The concepts of persistent and antipersistent memories in time are well-defined for non-linear processes (Feder, 1988).

The Hurst exponent not only works in the time-domain, but also in spatial domain to measure the roughness expressed in quantitative amounts (fractal dimensions and Hurst exponents). The interest in roughness studies has been motivated by Mandelbrot's work, in which he was faced with "The challenge to explain why so many rough facets of Nature are scale-invariant". In particular, the fault roughness can be studied using a fractal analysis (by means of the Hurst exponent or fractal dimension), as presented by Power et al. (1987); Schmittbuhl et al. (1993); Mandelbrot (2002). Also, the roughness of the magnitudes was calculated by means of the fractal image informatics toolbox (Oleschko et al., 2008). The roughness of earthquakes as a powerful tool to characterize the main features of seismicity and give insight into the inner dynamics of seismotectonic activity was studied by Telesca et al. (2001). For the magnitude estimations $M_w$, we created a firmagram as a plot for the discrete values of $M_w$ versus the Fault number. This compressed graph allows us to visualize the entire data density distribution and the peaks and valleys which are the result of irregularities and fluctuations in time series. The firmagram roughness for the three zones of the PAFS (western, central and eastern PAFS) can be measured, among others, by the Hurst exponent (see Fig. 3).

We used the Hurst exponent as the measure of roughness for the slip-rate series (time-domain) and for the magnitudes $M_w$ distributed along the PAFS (spatial domain). To estimate the Hurst Exponent we used the wavelet transform (Rehman and Siddiqi, 2009), wherein the characteristic measure is the wavelet exponent, $H_w$ (Malamud and Turcotte, 1999).

### 4.3 Wavelet variance analysis

The wavelet transform, introduced by Grossmann and Morlet (1984), is a filter function which is passed over time series and provides information on both space and frequency domains. A family of wavelets can be constructed from a function known as a "mother wavelet," which is confined in a finite interval. Then, "daughter wavelets" are formed by translation and contraction. The transform has a fractal basis, and the variance of wavelets obeys a power law, from which you can calculate the fractal dimension. In general, *wavelet variance analysis* is the most satisfactory measure of the persistence or antipersistence strength when only a small number of samples are available (Simonsen et al., 1998). We consequently selected this method because there are but few available slip-rate samples along the PAFS.

### 4.4 Box Dimension

The box counting method is the standard technique to prove the fractal behavior (scale invariance) and also a common way of estimating the fractal dimension. In order to obtain the fractal dimension of the slip-rate distribution for active faults, we

have used the *box counting 2D algorithm* (Walsh and Watterson, 1993), to obtain the box-dimension ($D_b$) in the following relationship:

$$N(e) \approx \frac{1}{e^{D_b}}, \tag{1}$$

where $N(e)$ is the number of boxes of linear size $e$ necessary to cover a data set of points or objects distributed on a two-dimensional plane. The basis of this technique is to measure $D_b$ by counting the boxes containing at least one point of the structure. Figure 4b plots the logarithm of $N(e)$ on the vertical axis versus the logarithm of $e$ on the horizontal axis. If the set is indeed fractal, this plot will follow a straight line with a negative slope that equals $-D_b$. For the box dimension calculation, we used the Benoit 1.3 Software (Trusoft, 1999).

### 4.5 Variograms

Commonly, natural phenomena exhibit anisotropic behavior such as seismic anisotropy, mineral veins, groundwater contaminant plumes, porosity, permeability, and other petrophysical characteristics, where the studied properties depend on the direction. In practice, variograms are studied in different directions to determine the presence or absence of anisotropy. Mathematically, a variogram represents the semivariance of data as a function of the distance that separates a pair of observations (Journel and Huijbregts, 1978). Generally, it is a function which increases with the distance and is canceled when the distance equals zero.

The spatial structure and anisotropy are revealed by the variogram surface. If the pattern forms an elliptic shape, it indicates the direction of best and poor correlations. Once the anisotropic feature is identified (often presented by an angle spectrum), the directional variograms are computed. A directional variogram can be obtained by calculating variogram at different distances and angles. The isotropic variogram obtained must exhibits a very good spatial structure and it is fitted with theoretical model, such as spherical or exponential functions. We used this variogram analysis to obtain the preferential direction of the faults that we propose as active, and to prove that these structures are optimally oriented in relation to the current stress field in the central TMVB ($\sigma 2$=NE-SW; see Fig. 5).

### 4.6 Intensity Scale (ESI 07)

The Environmental Seismic Intensity scale ESI 07 (Michetti et al., 2007) is a new intensity scale, with 12 degrees of intensity, based only on Earthquake Environmental Effects (EEEs). The ESI 07 scale integrates traditional intensity scales, and allows to define seismic intensity based on the entire scenario of effects. According to the ESI 07, all the paleoseismologically investigated faults of the PAFS are capable of generating Class B events (assessment of seismic intensity levels IX to X), with frequent and characteristic geomorphological and geological evidence.

## 4.7 Active Fault Definition

Finally, this work contributes with an intrinsic definition of active fault sources within the PAFS. Active Faults are those that are ground-rupturing with slip rates of approximately $0.001 \ \text{mm yr}^{-1}$, with associated seismic activity, at least in the last 10,000 yr, and are optimally oriented in relation to the current stress field (see Focal Mechanisms, slip rates of coseismic faults and semivariograms). The active fault planes must be related to earthquakes of a minimum magnitude of $M_w \geq 5.5$, or capable of generating rupture lengths greater than or equal to 3 km. If the active fault presents seismicity with these characteristics, it will be considered a seismogenetic fault.

## 5 Results

To assess the impact that moderate to strong seismic events can cause on cities in central Mexico, we defined the intracontinental faults capable of generating moderate to strong earthquakes by incorporating quantitative parameters (fractal dimension, Hurst exponent, and anisotropy). In this paper, we examined persistence on slip-rate time series and roughness on $M_w$ series. We can manage these statistical techniques because this fault population presents a self-similar behavior. This means that the log-log plot of frequency versus lengths for the PAFS obeys an inverse power law (distribution on a straight line), but is characterized by a bimodal self-similar scaling law with two slope values. This bimodality may reveal the existence of at least two different fracture processes in the fault system. A key step in this study was to delineate different zones of deformation processes using the temporal/spatial Hurst analysis. The results in the spatial domain strongly suggest that the PAFS is classified in three different zones (western PAFS, central PAFS and eastern PAFS) in terms of their roughness ($H_w = 0.7$, $H_w = 0.5$, and $H_w = 0.8$ respectively; Fig. 3), with their corresponding magnitudes ($5.5 \leq M_w \leq 6.9$; $5.5 \leq M_w \leq 6.7$; $5.5 \leq M_w \leq 7.0$ ). For the time domain, with a strong persistence of $H_w = 0.949$, the result suggests that the periodicities of slip-rates are close in time. The fractal capacity dimension ($D_b$) is also estimated for the slip-rate series. We found that $D_b = 1.86$ is related to a lesser concentration of low slip rates in the PAFS, suggesting that larger faults accommodate the strain more efficiently (length $\geq 3$ km).

We can prove, in terms of fractal analysis, that the 316 faults studied for the PAFS are seismically active. And in terms of variogram analysis, an anisotropic direction was identified in $80°$ ENE direction (Fig. 5), so these faults are optimally oriented in relation to the current stress. Moreover, they can generate average maximum and minimum magnitudes between $5.5 \leq M_w \leq 7$, which according to the Environmental Seismic Intensity scale ESI 07 correspond to a wide affected area (1 000 $\leq \text{km}^2 \leq 5$ 000). The size of this area means that movements in any of the PAFS faults would affect some of the most populated cities of central Mexico, such as Mexico City (population $\sim 9 \times 10^6$), Ecatepec ($\sim 1$ 600 000), Toluca (>800 000), Acambaro (>100 000), Maravatío (>80 000), Zinapécuaro (>50 000), Morelia ($> 10^6$), Pátzcuaro ($\sim 80$ 000), among others.

## 6 Discussion

In order to characterize the seismic potential of the PAFS, the analysis of the three empirical relations results of active faults was summarized as follows: (1) The model proposed by Anderson et al. (1996) always yields lower results than the other relationships; however, (2) the highest magnitudes are obtained with the relationship of Wesnousky (2008); (3) the average magnitudes are obtained with Wells and Coppersmith (1994). We have observed that all three relationships work for the PAFS. However, in this paper, we reported the maximum and minimum earthquake magnitudes estimated by Wells and Coppersmith (1994), because this method is best suited for areas with crustal thickness $> 15$ km and avoids overestimating the magnitudes.

The $M_w$ distribution organized from east to west is detailed on the Firmagram (Fig. 3). Moreover, we can observe the variability of the Hurst exponent along the PAFS, from $H = 0.8$ (eastern zone), $H = 0.5$ (central zone) to $H = 0.7$ (western zone). The result strongly suggests a micro-regionalization of the PAFS into three main zones. This micro-regionalization is of paramount importance in seismic hazard analysis and in the understanding of fault dynamics. The persistence values ($H > 0.5$) are related to the predictability of future seismic events based on the existing correlation with past events. They are widely consistent with the instrumental seismicity, because the eastern zone is the most active, followed by the western segment of the PAFS. Meanwhile, the central PAFS corresponds to a random process. As a consequence, there is a dependence and causality between $H$ and the PAFS dynamics. The differences between each zone are:

- **Eastern PAFS:** This zone is the most active sector, based on $H = 0.8$ and the obtained magnitudes of $5.5 \leq M_w \leq 7.0$. This is evidenced by the paleoseismological studies and the instrumental seismicity. Regarding persistence, these earthquake magnitudes are susceptible to ground-rupturing, showing an increasing trend towards the future. Coupled with the results of (**a**) Velázquez-Bucio (2018), it appears that several segments of the Acambay graben are already at their time zero and could break any time and (**b**) Arzate et al. (2018) found that the central graben fault system converge at a depth of $\sim 18$ km, that fact represents the possibility of occurrence of earthquakes rupturing along various faults of the Acambay graben as observed for the 1912 earthquake ( Langridge et al., 2000; Suter et al., 2015 and Ortuño et al., 2018) . This configures an area with high seismic hazard.

- **Central PAFS:** $H = 0.5$ shows a Brownian process, no trend, for magnitudes between $5.5 \leq M_w \leq 6.7$. We suggest here that $H$ could be related to shorter fault lengths, and consequently there is a lesser amount of deformation in the area. This result can be related to the emplacement of the Los Azufres Geothermal Field and its magmatic chamber, located between 4.3 and 9.5 km of depth at the El Guangoche dome. Moreover, the E-W Pátzcuaro-Acambay Fault System is affecting the 1 Ma rhyolitic and dacitic domes (Agua Fría), but also the andesitic volcanism, and controlling the distribution of monogenetic volcanism in the area. So, the observed value of $H$ depends, among other things, on the fragile-ductile limit of the crust. All these facts allow to validate that the central zone differs tectonically from the eastern and western sectors, and is characterized by a seismic gap similar to that of the Tzitzio-Valle de Santiago fault.

- **Western PAFS:** Persistent values of $H \sim 0.7$ have been reported by other authors (e. g. Scholz, 1997; Schmittbuhl et al., 2006). In our case, this value is consistent with the persistence of earthquake magnitudes ranging between $5.5 \leq M_w \leq$

6.9 (faulting processes with memory). In fact, the paleoseismological studies by Garduño-Monroy et al. (2009) and Soria-Caballero (submitted), indicated similar magnitudes for historic earthquakes in the zones of Zacapu, Pátzcuaro, Morelia, and Cuitzeo. Therefore, the western PAFS is a high seismic hazard zone too, but to a lesser degree than the eastern zone.

For the slip-rate time series we reported values of $H = 0.949$ and $D_b = 1.86$. The Hurst exponent shows a strong persistence, meaning close periodicities in time for ground-rupturing in the PAFS. $D_b = 1.86$ is consistent with the values obtained by Nieto-Samaniego et al., (2005; $D_b = 1.87$ upper limit), who proved that the box dimension is in inverse relation to fracture concentration and in direct relation to fracture density. The high value of the fractal dimension might indicate the possibility of a major earthquake (Aviles et al., 1987) in the PAFS faults. The high $D_b = 1.86$ value obtained depends inversely on the failure concentration criterion, which indicates that the critical fault concentration, based on the stability of two faults in a stress field, is directly proportional to the factor $(L/l)$, where $L$ is the size of the region and $l$ the mean fault length (Smirnov and Zavyalov, 1997). So, low critical fault concentration corresponds to short fault lengths: it is well-known that lesser amounts of deformation are directly proportional to short faults lengths and low slip rates. Therefore, short faults lengths in the PAFS accommodate little deformation, suggesting that fault lengths of $\geq 3$ km accommodate the deformation of the PAFS more efficiently.

Finally, supported by $D_b$ and $H_w$, we can neatly determine the lower limit (3 km) of fault lengths for the PAFS. However, we cannot establish a definite upper limit due the faults hidden under Holocene deposits, not identifiable on a 15 m Digital Elevation. We nevertheless estimated an upper limit of fault lengths (38 km) as a first approximation.

## 7 Conclusions

Spatial-temporal methods were applied to fault data, and the fractal behavior observed for the entire PAFS allowed to define which segments can be designated as active faults. Therefore, an active fault is defined as a plane that presents ground-rupturing with slip rates of approximately 0.001 mm yr$^{-1}$, and associated seismic activity at least in the last 10,000 yr. Moreover, active faults are optimally oriented to the current stress field, and the active fault planes must be related to earthquakes with a minimum magnitude of $M_w \geq 5.5$, or capable of generating rupture lengths greater than or equal to 3 km. The temporal slip-rate distribution for the PAFS displays a fractal behavior, with strong persistent characteristics ($H = 0.949$). In this context, we have a statistical measure of the memory of slip-rate series, and we can infer the predictability of these temporal series, to conclude that the entire fault system is active. Moreover, we were able to define a micro-regionalization of the PAFS using the relationship established between $H$ and the PAFS dynamics. $H = 0.5$ for $M_w$ behaves like a brownian process,(Central PAFS), and $H > 0.5$ for $M_w$ has a trend (easter PAFS and western PAFS). The result reveals the easter PAFS as the most active zone. With regard to the regional structures and their relationships with magmatism or hydrothermal processes, it is clear that there are three zones within the PAFS, where faults with different geometries and also different magmatic processes were observed, surely related to the values of $H_w$ obtained for each zone. In particular, the estimation of maximum and minimum earthquake magnitudes ($5.5 \leq M_w \leq 7.0$) is likely to affect a large area ($1\,000 \leq$ km$^2 \leq 5\,000$) in the central region of Mexico, where there

are many cities with high population density. As discussed earlier, assessing seismic hazard and improving of the vulnerability studies is acutely necessary in the central portion of Mexico. Finally, we conclude that the PAFS fulfills the intrinsic definition of active fault. The PAFS is likely to cause future social concern. As such, we strongly believe that the area must continue to be investigated by multidisciplinary studies in order to improve territorial planning and reduce seismic hazard in central Mexico.

*Data availability.* The datasets generated during the current study are available from the corresponding author on reasonable request.

*Author contributions.* AMP built the GIS-based database of active faults, supervised by VGM and DSC. AMP and AFS performed the fractal analysis and the estimations of the maximum magnitudes, as well as the discussion of the results. AMP and AFS took the lead in writing the manuscript. DCS contributed with the Intensity Scale values for the active faults in the PAFS and together with VGM helped shape the research. VGM also performed the review of the tectonics in the PAFS and helped supervise the project. All authors discussed the results and
10 contributed to the final manuscript

*Competing interests.* The author declares no competing financial interests

*Acknowledgements.* SSN data was obtained by the Servicio Sismológico Nacional (México), station maintenance, data acquisition and distribution is thanks to its personnel. This work was partially funded by the P17-CeMIEGeo and PT5.2 GEMex projects of the Universidad Michoacana de San Nicolás de Hidalgo, and by CONACYT scholarship No. 234243 (A. Mendoza-Ponce). We really thank all the referees
who contributed for their time, effort, and suggestions to making a better manuscript.

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

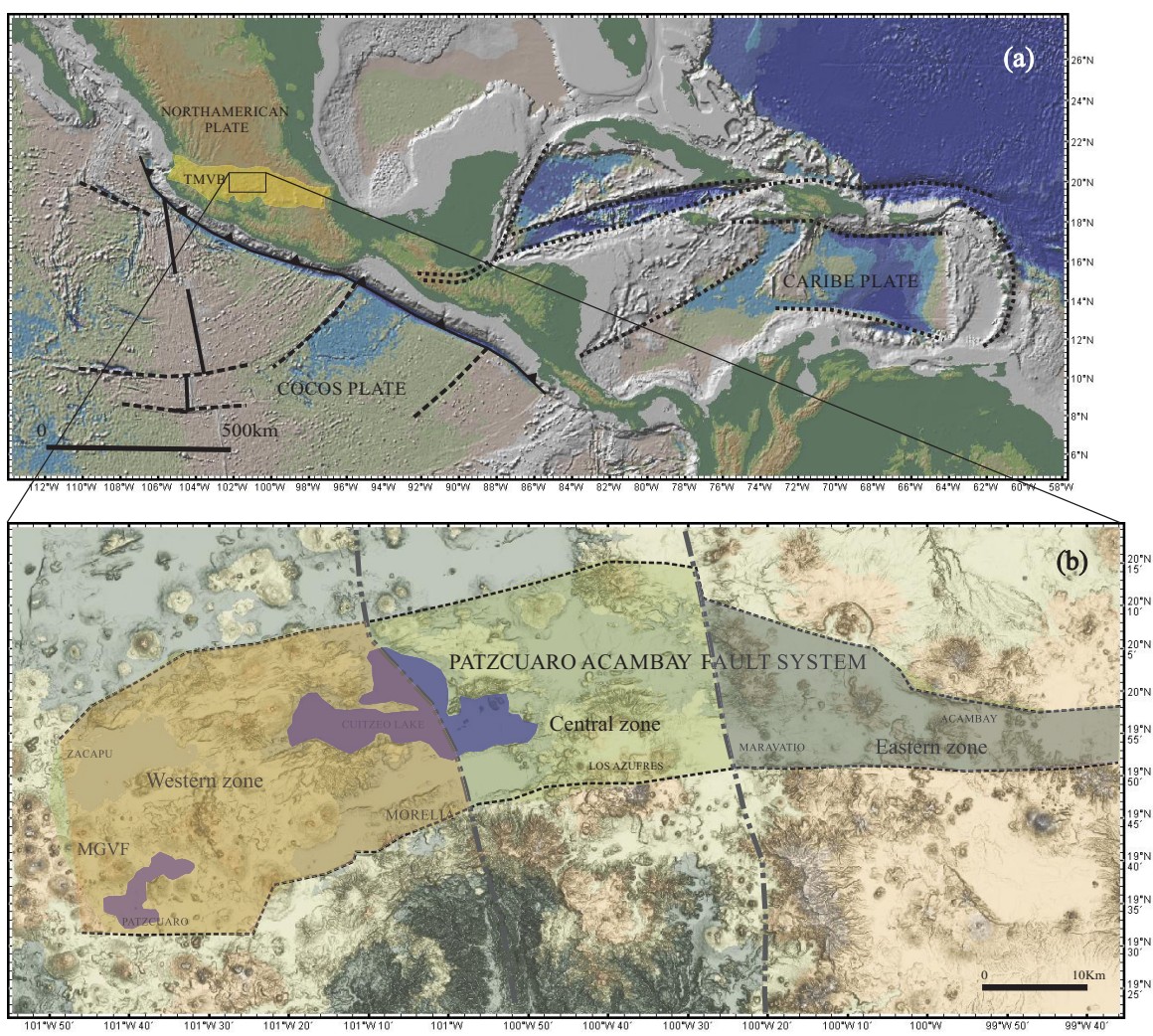

**Figure 1. (a)** Tectonics and geodynamic setting of the Trans-Mexican Volcanic Belt (yellow area). Dotted black lines are plate boundaries and the continuous black line is the Middle America Trench. The Pátzcuaro-Acambay Fault System (PAFS) is outlined by the black rectangle. TMVB: Trans-Mexican Volcanic Belt. **(b)** Tectonic area of the Pátzcuaro-Acambay Fault System that shows the limits (thick dashed gray lines) between the three zones of the PAFS (western, central and eastern zones). MGVF: Michoacán-Guanajuato Volcanic Field.

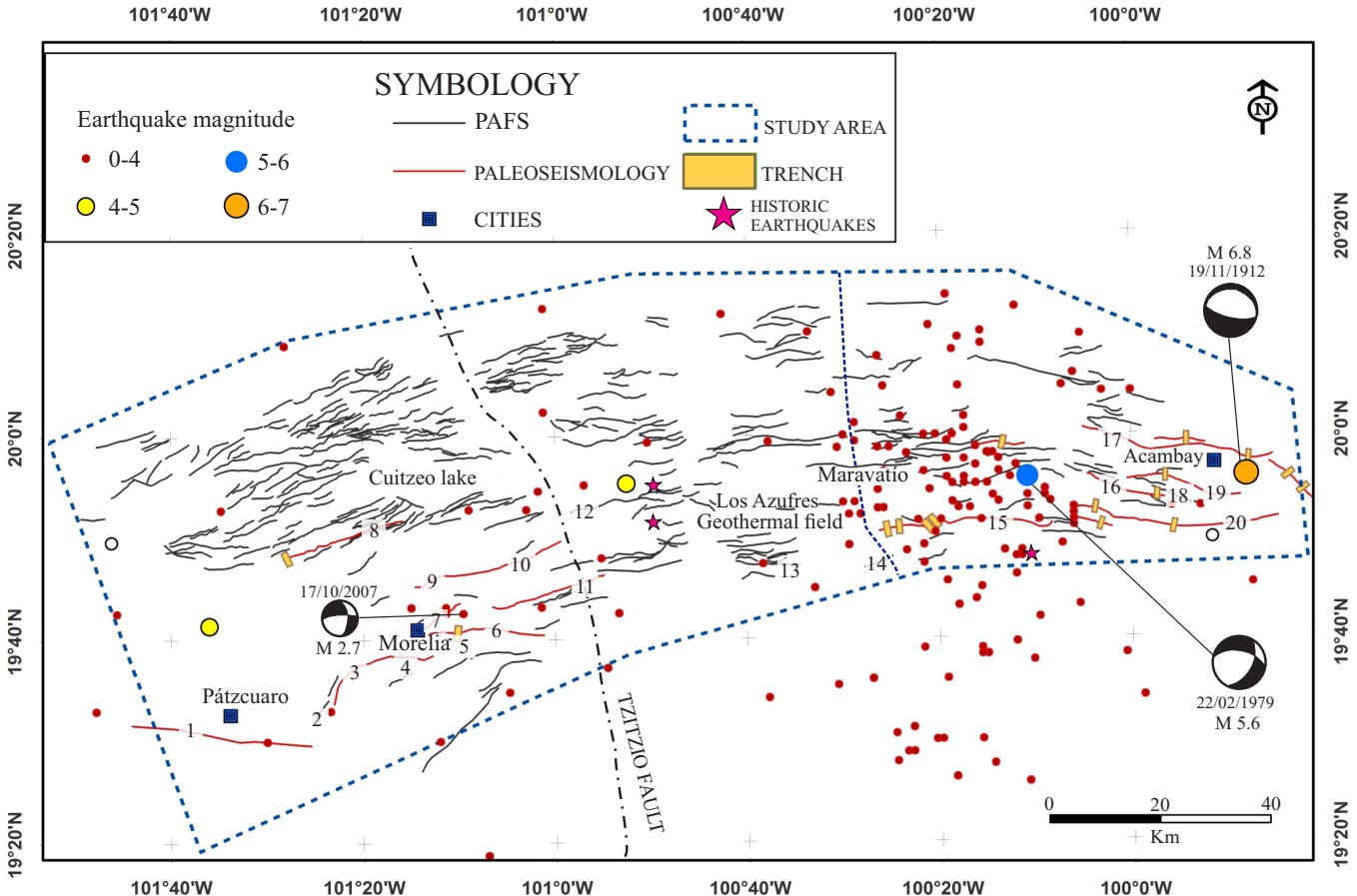

**Figure 2.** Structural map of the Pátzcuaro-Acambay Fault System. Active Faults are represented by continuous black lines. Continuous red lines are the faults studied with paleoseismological approach. The purple dotted line is the limit between the central and eastern zones and the Tzitzio Fault is the limit between the western and central zones. Stars represent the seismic crisis in Pátzcuaro and Araró (in 1845 and 1858) and Zinapécuaro and Tlalpujahua (in the 19th century). Circles represent the seismicity from 1912 to 2018 based on the catalog of the National Seismological Service of Mexico (Servicio Sismológico Nacional, SSN). Focal Mechanisms were reported by Astiz-Delgado (1980), Suter et al. (1992; 1995), Langridge et al. (2000), Singh et al. (2011; 2012), and Rodríguez-Pascua et al. (2012). The ascending numbers are referred to in Table 1.

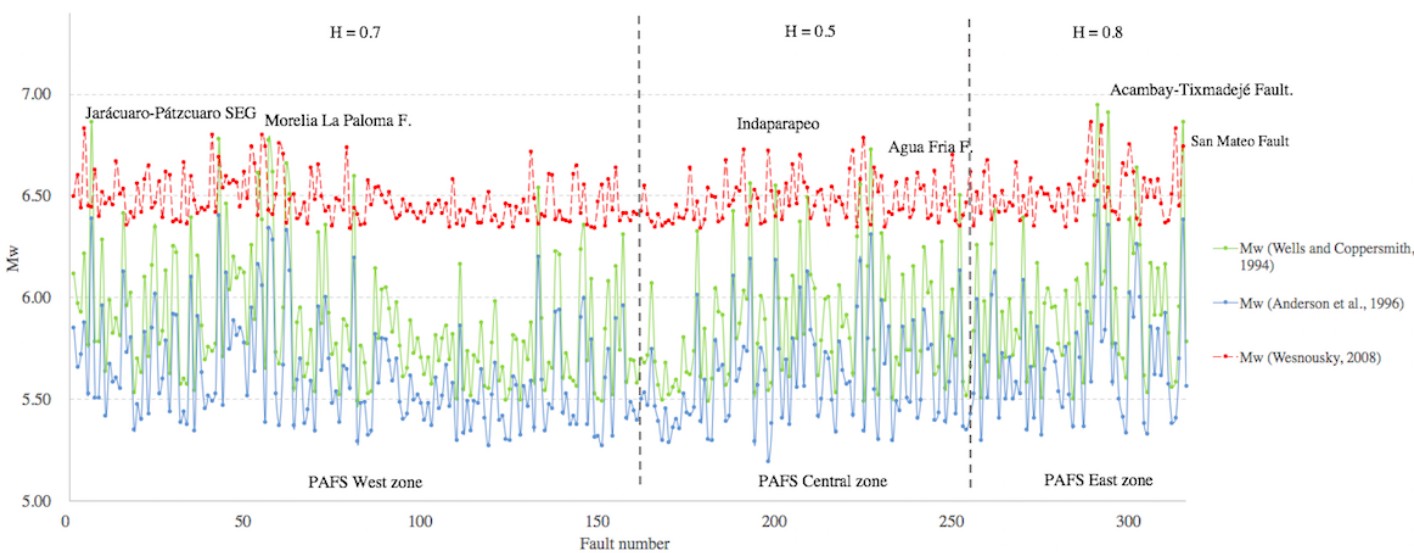

**Figure 3.** Firmagram roughness graph for the earthquake magnitude variations calculated by the surface rupture length (SRL) for the PAFS using: Wesnousky (2008, data in red); Wells and Coppersmith (1994, data in green) and Anderson et al. (1996, data in blue). By optical visualization we observed that Wells and Coppersmith (1994) and Anderson et al. (1996) are underestimating the data with respect to the magnitudes determined by the equivalent regression model proposed by Wesnowsky (2008). Dashed gray lines are delimiting the three zones that define the micro-regionalization of the PAFS (western PAFS , central PAFS and eastern PAFS ) in terms of their Hurst exponent ($H_w = 0.7, H_w = 0.5, H_w = 0.8$ respectively), showing different seismic rates for each zone. Some fault names were printed in the figure.

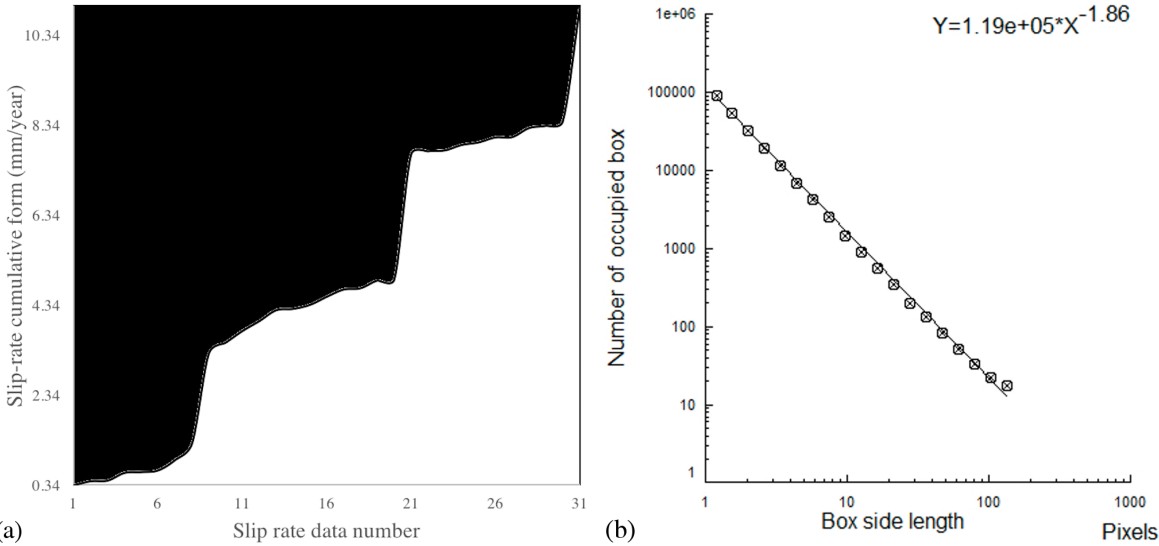

**Figure 4. (a)** Devil's staircase fractal for the cumulative slip-rate (mm/yr) distribution of active faults from the east to west in the PAFS (see Table 1). **(b)** Log-log plot of number of boxes ($N(e)$) versus box side length ($e$). The slope of the straight line is equal to the capacity dimension or box-dimension. The resulting $D_b = 1.86$ is related to a lower concentration of low slip rates in the PAFS, suggesting that larger faults accommodate the strain more efficiently.

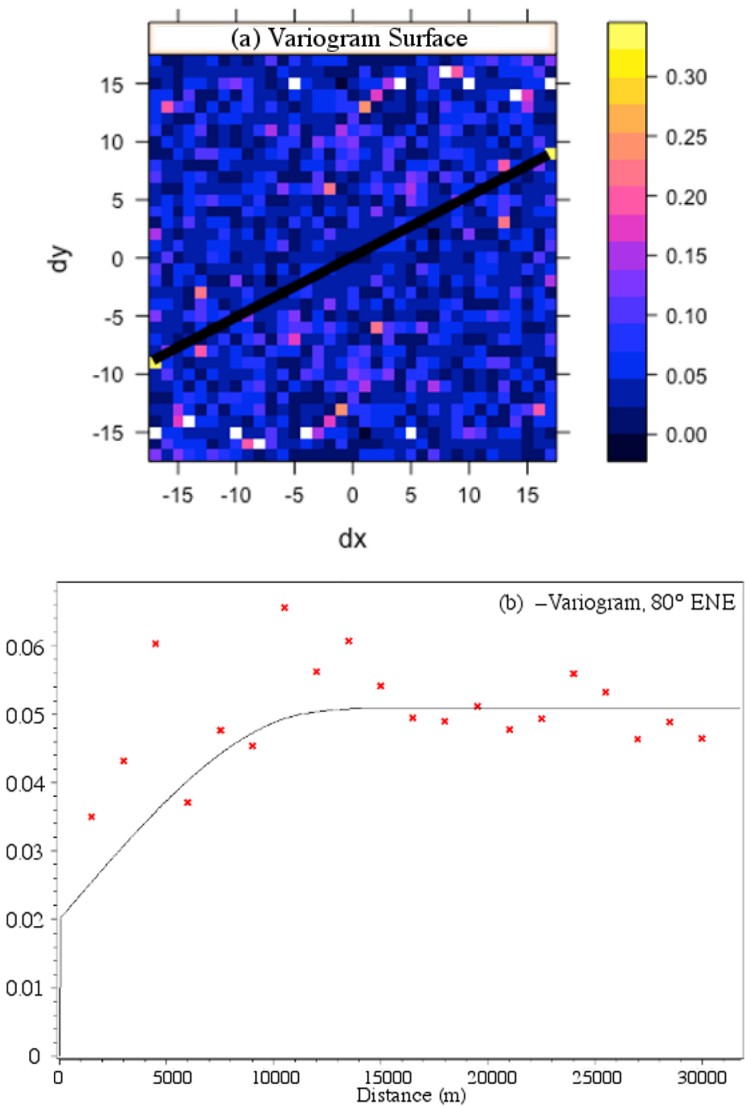

**Figure 5. (a)** Variogram surface of fault lengths in the PAFS, with a straight line showing the direction of the best spatial autocorrelation. At the center a pattern forms a shape of an ellipse, suggesting that the best correlations are observed in NE-SW direction. **(b)** Anisotropic variogram of fault lengths, which exhibits a good structure in ENE direction (80°), and matches with a spherical model (black continuous line). So, this faults are optimally oriented in relation to the current stress in the PAFS ($\sigma 2$). So, this anisotropic direction indicates that the spatial distribution of faults is mainly due to extensional stresses, and is consequently subject to deformation generated by the current stress field in the PAFS ($\sigma 2$, NW-SE).

**Table 1.** Slip-rate estimations of earlier studies organized from east to west along the PAFS.

| Fault name | Length (km) | Scarp (m) | Slip-rate (mm/yr) | Slip-rate cumulative-form | References | $M_w$ estimated (W. and C., 1994) | ESI 07 scale | Affected area (km$^2$) |
|---|---|---|---|---|---|---|---|---|
| (17)Colluvium 60* | | | 0.17 | | Suter et al. (2001) | | | |
| (17)Acambay-Tixmadejé | 36 | 650 | 0.17 ± 0.02 | 0.34 | Langridge et al. (2000) | 6.91 | X | 5000 |
| (19)San Mateo | 21 | 115 | 0.085 ± 0.025 | 0.43 | Sunye-Puchol et al. (2015) | 6.62 | IX | 1000 |
| (18)San Pedro Alto | 6 | 136 | 0.0159 | 0.44 | Velazquez-Bucio (2018) | 6.17 | IX | 1000 |
| (16)Temascalcingo | 15 | 230 | 0.17 | 0.61 | Ortuño et al. (2018) | 6.41 | IX | 1000 |
| (16)Temascalcingo | 15 | 230 | 0.0173 | 0.63 | Velazquez-Bucio (2018) | 6.41 | IX | 1000 |
| (20)Unnamed Basalt 59* | 15 | | 0.04 | 0.67 | Suter et al. (2001) | | | |
| (20)Pastores (P) | 38 | 243 | 0.23 | 0.90 | Ortuño et al. (2015) | 6.95 | X | 5000 |
| | | | 0.37 | 1.27 | | | | |
| (15)Venta de Bravo | 33 | 300 | 2 | 3.27 | Suter et al. (1992) | 6.86 | X | 5000 |
| (15)Venta de Bravo | 32 | 300 | 0.24 | 3.51 | Ortuño et al. (2018) | 6.86 | X | 5000 |
| | | | 0.26 | 3.77 | | | | |
| (15)Venta de Bravo | 47 | 50-300 | 0.22 | 3.99 | Lacan et al. (2018) | 6.86 | X | 5000 |
| | | | 0.24 | 4.23 | | | | |
| (14)Cd. Hidalgo Basalt 65* | | 20 | 0.03 | 4.26 | Suter et al. (2001) | | | |
| (14)Cd. Hidalgo Basalt 47* | | 70 | 0.09 | 4.35 | Suter et al. (2001) | | | |
| (13)San Andrés Dacite 50* | | 60 | 0.18 | 4.53 | Suter et al. (2001) | | | |
| (12)Cuitzeo Basalt 37* | | 120 | 0.16 | 4.69 | Suter et al. (2001) | | | |
| (12)Cuitzeo Basalt 39* | | 20 | 0.03 | 4.72 | Suter et al. (2001) | | | |
| (11)Charo-Queréndaro | 21 | 80 | 0.17 | 4.89 | Garduño-Monroy et al.(2009) | 6.62 | IX | 1000 |
| (10)Tarimbaro-Á. Obregón | 28 | 200 | 0.025 | 4.91 | Garduño-Monroy et al.(2009) | 6.78 | X | 5000 |
| | | | 2.78 | 7.69 | | | | |
| (9)Quinceo Basalt 41* | | 40 | 0.07 | 7.76 | Suter et al. (2001) | | | |
| (9)Quinceo Basalt 44* | | 10 | 0.02 | 7.78 | Suter et al. (2001) | | | |
| (7)Morelia C. Camionera | 10 | 60 | 0.12 | 7.90 | Garduño-Monroy et al. (2009) | 6.15 | IX | 1000 |
| (6)Morelia La Paloma | 21 | 300 | 0.057 | 7.96 | Garduño-Monroy et al. (2009) | 6.61 | IX | 1000 |
| (8)Teremendo | 23 | 70 | 0.11 | 8.07 | Soria-Caballero (Submitted) | 6.66 | IX | 1000 |
| (5)Morelos | 8 | 50 | 0.009 | 8.08 | Garduño-Monroy et al.(2009) | 6.03 | IX | 1000 |
| (4)Cointzio | 14 | 100 | 0.2 | 8.28 | Garduño-Monroy et al.(2009) | 6.38 | IX | 1000 |
| (3)C. El Aguila lava 42* | | 40 | 0.05 | 8.33 | Suter et al. (2001) | | | |
| (2)Huiramba | 5 | 50 | 0.1 | 8.43 | Garduño-Monroy et al.(2009) | 5.74 | IX | 1000 |
| (1)Jarácuaro-Pátzcuaro | 33 | 277 | 2.5 | 10.93 | Garduño-Monroy et al.(2009) | 6.86 | X | 5000 |

*Scarp refers to the top of the faulted rock unit.

Here, we reported the maximum earthquake magnitudes obtained with Wells and Coppersmith (1994). The ascending numbers are referred to in Fig. 2.