# Peer review of "Active Faults Sources for the Pátzcuaro-Acambay Fault System (Mexico): Fractal Analysis of Slip Rates, and Magnitudes $M_w$ Estimated from Fault Length."

_Natural Hazards and Earth System Sciences, 2018_

## Referee Comment (RC1) · P. Lacan (Referee) · 13 Apr 2018

Avith Mendoza-Ponce and co-workers manuscript represent a potentially interesting contribution to the characterisation of active faulting along the Morelia-Acambay Fault System in Mexico.

Hundreds of demonstrated and presumed active faults that form the Morelia-Acambay fault system have been mapped and different parameters were used to propose a characterization of the entire fault system.

[Figure]

The manuscript presents potentially interesting data but suffers from a poor organization and writing that make it very difficult to follow. In my opinion, the data recompiled are potentially of interest for the reader of NHESS but the manuscript must be greatly reorganized and rewritten, clarifying objectives and focusing on the new results (which are not easy to identify in the actual form). Below, I add major comments that could improve the manuscript:

1) General Organization:

The structure of paper is very confused and is not easy to find the elements to follow the reasoning of the authors. -The introduction should introduce the problematic of the manuscript by removing all the generalities away from the objectives of the paper. -The Seismotectonic Setting should be organized to set out the elements necessary for understanding the discussion. In its current form, everything is underneath and the state of art is not clear. -The methods should be explained more carefully: What morphological evidences have been taken into account delimiting fault segments and main faults. Why so much difference with works published previously. In particular Lacan et al., 2018 calculated 48 km length for the Venta de Bravo fault, you should explain how do you calculate the different length (32.982 km?) and why is this difference so important. Same for the Pastores Fault: 33 km for Langridge et al., 2013 and 38 km for you? and other faults. . . For the "2.4 Fractal analysis", you lose the reader with details explanations but you do not explain what you want to calculate? Why do you think it's fractal? What does these calculations represent? - The Result and Discussion part is confused. I strongly recommend separating the results from one side (explaining the results you get) and after, a discussion section where you discuss these results and their consequences. In the current form, we do not distinguish what is new from what was already known. In particular I do not understand the relationship between the results you present and the generation of major earthquakes. What is already known, including previous mapping of faulting in the area should be carefully presented in the seismotectonic setting. - The conclusion is also confused. You should clearly state

your results and their consequences on the seismic risk of the region.

2) Bibliography: You have probably had a problem with the bibliography editor, everything seems to mix and many references are not attributed to the correct concepts, ideas or interpretation.

3) Figures:

Figure 1: "Geodynamic" and "Seismotectonic setting" imply to indicate the geodynamic elements, the seismicity and the main tectonic features on the figure. Figure 2: It is absolutely necessary to explain in the text how did you realize the fault mapping. Why, for example, some faults located north of the figures are not drawn? What is the difference between your work and the cartography of Suter et al., 2001 or Ferrari et al., 2012? Figure 4: scale is missing

4) English must be revised

Given my comments and considering the importance of such a study for the understanding of the seismic risk in the central region of Mexico, I recommend major revisions before the article can be accepted for publication in NHESS.

---

## Referee Comment (RC2) · M. Meghraoui (Referee) · 17 Jun 2018

Comment on manuscript nhess-2018-63 titled "Active Faults sources of the Morelia-Acambay Fault System, Mexico based on Paleoseismology and the estimation of magnitude Mw from fault dimensions" submitted by Mendoza-Ponce et al.

The manuscript (ms) presents the fractal characteristic of active faults in the Morelia-Acambay Graben, their seismic parameters (length, width, slip, paleoseismic history) and fractal dimension and behavior. The issue is emphasized using a rich database

with numerous previously published neotectonic works. The statistical analysis is quite interesting in assigning a maximum magnitude and "area of influence" (earthquake damage area?) for the seismic hazard assessment. The article is however not well written and suffers of several weaknesses that make the presented work difficult to understand. I recommend a very major revision.

Here are some recommended general and specific changes that may improve the presentation of the manuscript:

General remarks ïĆǧ The main topic of the ms is on the fractal fault distribution and its related seismic activity but this is not clear neither from the title, nor for the abstract and text. This article needs to be restructured in order to clearly put forward the fractal analysis, the authors do not present new fault data and hence, the presented neotectonic and seismotectonic characteristics cannot be considered as the main topic of this article. ïĆǧ The authors mention the existence of 316 fault segments in text and about 22 fault characteristics (in Table 1) of the Morelia Acambay Graben. However, they do not explain how they did select these 22 items among the 316 faults, and which fault segments where used for the fractal analysis. The 316 fault segments deserve to be shown as a supplemental material. ïĆǧ The seismicity and neotectonic database and related catalogs need to be clearly presented in the form of tables with appropriate legends showing the origin of data. A table of paleoseismic, historical and instrumental earthquakes is needed in this manuscript, at least for earthquakes with Mw $\geq$ 5.4 (according to their concluding remarks). Table 1 needs a to include the minimum and maximum, observed and estimated coseismic slip/event for the known faults. Table 1 needs a serious legend. ïĆǧ An interesting issue is the difference between the fracture density and fracture concentration. This section of the manuscript needs to be developed in order to show the meaning of this difference, explain well the correlation between box dimensions and the effects of the size of fracture concentration. The calculation of the Hurst Exponent H and related strong persistent process, Devil staircase and box dimension should be explained more extensively. These aspects that

are fundamental in this manuscript should appear in a separated Methodological section. ïĆǧ The uncertainties of seismic and neotectonic data are totally neglected in this manuscript.

Specific remarks Title: It has to be reconsidered because as presented, it shows that active faults and paleoseismic analysis are the main topic of the manuscript. I think that the fractal analysis from existing fault data should be clearly announced in this title.

Abstract: The authors use different magnitude scales (Ms, Mb, Mw). If a seismicity catalogue with homogenized magnitudes exists for Mexico, then the authors should use Mw only in this section. The main addressed topic (fractal analysis) that is left in the last 4 or 5 lines of the abstract should be put forward.

The Introduction section is not well written, and although it includes several paragraphs as seismotectonic settings, it does not explain the geodynamic context with clear stress and strain distribution. For instance, Figures 1 A and B that are redundant they show only the topography and bathymetry. Figure 1C is supposed to show the seismotectonic setting but it looks only like a geographic indication of the Morelia-Acambay Graben. The introduction needs to be better organized to explain the context and main issue, the used general methodology (fractal analysis) and its application elsewhere in comparable seismotectonic domains, previous works emphasizing the main results and finally the main steps adopted in this ms.

(Neotectonic and seismotectonic settings?) Since the Morelia-Acambay Graben has a rich database, a specific section in neotectonics and seismotectonics would therefore be needed after the introduction. In this case, the authors should organize their text and avoid a mix of data. This section needs to present: 1) the seismicity (historical and instrumental) with emphasis on major events and their characteristics, 2) the geodetic results (GPS, conventional), focal mechanism solutions and fault kinematics for the stress and strain distribution, and 3) the paleoseismic data and results including the estimated slip rates with the corresponding time window and related uncertainties. This

section has not to be long but it has to focus on major results showing the related references and how completed is the database (reference to tables in supplementary material is recommendable).

Line 20 – 21: Please note that historical earthquakes needs to indicated with their intensities (or inferred magnitudes), their severity (number of victims whenever possible). Line 22: ".. set of earthquakes . . ." of what magnitudes? Line 26_27: These lines are concluding remarks and should be moved at the end of ms.

Line 40: Instead of cortical, the term "crustal" is ususally used in active tectonics. Line 46: "The kinematics of them . . ." change in Their kinematics . . . This sentence mentions details on the neotectonic episodes and a reference is needed here. Line 49: normal-right ? change in Oblique fault with right-lateral normal component. Line 53-54: You give Dmax to all faults except to the Pastores Fault, why? Line 55: The 8.2 km depth of the Maravatio earthquake needs uncertainties. The sentence should be rewritten "Subsequently, another earthquake in 1979 with a magnitude Mb = 5.3 and a depth of 8.2 km (Astiz-Delgado, 1980), caused major damage in Maravatío." Line 59: "is very probable that this sequence of earthquakes is related to the La Paloma fault of 13 km of length . . .". How did you infer this? If this is obtained from the two local stations then the "probable" should turn into "possible". Please explain. Line 59-60: ". . . active from the Holocene" does not mean much. I would suggest considered active because it affects Holocene deposits. Line 63: remove seismic risk and put seismic hazard instead. Lines 65 to 84: In all these paragraphs, slip rates need to be explained (from which field trenches and markers, e.g., lateral or vertical offset of streams, . . .) and measurements span which timeframe. Line 81: What is the mechanism of the dozen faults? Are they in table 1.

Line 90: Active faults are . . . Lin 92: ". . . speeds of approximately . . ." ; fault speed is not used in active tectonics. Slip rate is more appropriate. Please apply correction throughout the text. Line 94 : . . . or capable of generating coseismic rupture length . . . How about coseismic displacement (slip)?

Line 91: The title is inappropriate in this ms. You are only extracting the data from previous works and not mapping and describing the faults of the Morelia-Acambay Graben.

Figure 2 is a bad quality map. Unless a clear srtm background topography can be shown, it should be removed, leaving only the seismicity and tectonic data in the map. The dates and magnitudes of focal mechanisms need to indicated in the map and in a table with their characteristics (in the supplemental material).

Line 100: CeMIEGeo - Project 17 needs a reference. Line 102: Unless you indicate criteria for selection, the characteristics of the 316 fault segments need to be shown at least in the supplemental material.

Lines 105 and 106: Fault length, Fault scarp height (?). Line 111: Distance between a locality and fault zone.

Lines 115 to 120: The use of the empirical equations (Wells and Coppersmith, 1994; Anderson et al. 1996; and Wesnousky, 2008) is a solution for the Mw determination. However, there is also another method using simply the seismic moment Mo = $\mu$SU (as defined by Aki (1967). In this case, you will better estimate your moment magnitude (Mw from Hanks and Kanamori, 1979) taking into account the uncertainties of fault parameters (length, width and average slip U). Estimation of Mw magnitudes as shown in Figures 3 a and b needs a reevaluation. Including the uncertainties of fault parameters is critical in the fractal analysis.

Line 126: The reference of Hurst (1951) for the Hurst component for the roughness measurement is needed here.

The section 2.4 on the fractal analysis is devoted almost entirely to the methodological aspect; please indicate it accordingly as for instance "Method of faulting study using fractal analysis". The manuscript is mainly based on this methodology section and it should be presented before the database (seismotectonic) section.

Line 149: In equation (??), please complete.

Line 162: . . . as fault planes . . . Also remove speeds, and replace by slip rate. Line 164-165: " . . . earthquakes of magnitude Mw $\geq$ 5.2 or related to rupture lengths greater than or equal to 3 km." Why Mw $\geq$ 5.2 and why lengths $\geq$ 3 km? How about hidden faults below Holocene deposits? As indicated by Langridge et al., (2013) and Sunye-Puchol et al., (2015) some faults can be hidden by young sedimentary deposits. In this case the fault lengths may increase. This issue needs to be discussed.

Line 174-175: Recurrence interval of which earthquake magnitudes? Line 177-178: The described seismicity, frequency and related b-value which is also a fractal distribution needs to be called earlier along with the fractal analysis in this manuscript. As this work is based on the Magana-Garcia Master thesis, that is not published and difficult to access as a reference, it should be presented with some details in introduction and seismotectonic section (or even in the supplemental material).

Line 180: Why this Table 1 is called only in section 3. This reference to the database should be called earlier !!! Line 184: Please give a reference to the Environmental Seismic Intensity scale (ESI 07) Line 185: What are class B events? Line 189: Hurst (1951) does not exist on the list of references. Line 191-192: The reference to the Hurst Exponent H and strong persistent process for the slip-rate distribution, along explanations on the Devil staircase should be explained in the methodology section. Line 192-193: ". . .cycles or periods with different seismic activity . . .", you mean variable seismic cycles ? Line 195: Explanations on the Devil's staircase and related (very bad) figure 4 need a serious revision! Line 200: This has to be included in the Methodology section. Line 205: How do you determine the stability of faults? Line 207: The reference of Soria-Caballero et al., is missing eve if it is in preparation (please provide the manuscript). Line 221: What us the mathematical behaviour? You mean the mathematical or statistical expression of faulting behaviour?

Mustapha Meghraoui IPG Strasbourg, France

Please also note the supplement to this comment:
https://www.nat-hazards-earth-syst-sci-discuss.net/nhess-2018-63/nhess-2018-63-RC2-supplement.pdf

—————————————————

---

## Referee Comment (RC3) · Assist. Prof. Aksoy (Referee) · 30 Jun 2018

Overview of manuscript:

Mendoza-Ponce and co-authors present a work, where they aim to evaluate the characteristics of the seismic sources in the Morelia-Acambay fault system in Mexico by using a quantitative approach. Their approach considers the spatial properties and distribution of the active faults and slip-rate estimations of earlier studies. They analyse a data set of 316 faults, which is partly mapped by the authors and partly compiled from

earlier studies. The analysis involves fault lengths, distances between faults and their slip-rates. They calculate maximum earthquake magnitudes based on fault lengths. The statistical method Mendoza-Ponce et al. are using allows to distinguish random from non-random systems and to identify the persistence of a trend within a time series; here slip-rates (Hurst Exponent). According to their analysis, Mondoza-Ponce et al., conclude that 1) the entire faults system is active, 2) expected maximum earthquake magnitude for the study area is Mw 7.0 and 3) the studied faults are tending to growth. Consequently, they suggest that their findings contribute to define the seismic risk of the Morelia-Acambay Faults system and improve the vulnerability efforts in Mexico.

Remarks

This study offers a compilation (GIS database) for active faults with paleoseismic studies in the Morelia-Acambay Faults system Such data are useful in seismic hazard evaluation and therefore might be of interest to the readers of NHESS. On the other hand, the manuscript needs significant improvements on the presentation and structure of the work; more information methodology, the approach and the significance of the results. The text is very fluent and easy to read, however the manuscript is difficult to follow if a reader is unfamiliar to the study area and the applied methods. It provides a comprehensive summary for the seismotectonic setting of the area. However the figures lack significantly of useful information that are necessary to comprehend the study area. Many cities, locations, fault names mentioned in the text are not available in maps and figures, making it difficult for the reader to orient him/herself spatially. Although the authors provide some theoretical information on the statistical calculations the connection and relation to the seismic hazard evaluation is poorly given, the geological significance for each input and output are not provided and discussed in the manuscript sufficiently. The aim of the study is confusing because throughout the text authors describe several different purposes:

1-prepare an intrinsic definition for active faults (abstract) 2-estimation of possible maximum earthquake magnitudes (abstract) 3-understand the seismic activity from

Patzcuaro to Acambay sector (introduction) 4-define the intracontinental structures that are susceptible to generate moderate and strong seismic events (line 85)

Aside the quantitative results, the study addresses only the first two purposes clearly. Maximum earthquake magnitudes are calculated via fault length measurements and a comprehensive definition is given for active faults. Based on the Hurst Exponent it is concluded that the fault system is active, however the possibility of an inactive fault system is not discussed within the manuscript. The maximum magnitude proposed for the region is Mw 7.0 and is calculated using three different magnitude-scaling relationships. The authors illustrate (Fig 3) that these relationships give different magnitude estimations for the same fault section but do not discuss how they interpret this difference. No reasoning is provided why authors prefer to take into account the Wesnousky (2008) relationship. The analysis assumes that each fault section has the potential to rupture the entire crust individually; (at all scales like 3-5 km). Why is 3 km the minimum preferred fault length that is included into the dataset? Can these faults also create surface ruptures? Furthermore, this analysis needs to consider the spatial distribution and interaction of the faults. An earthquake may rupture several adjacent fault segments; which would necessarily imply a larger earthquake magnitude. Authors need to consider multi-segment ruptures according to fault segmentation patterns and spatial distribution of the faults. Therefore, I consider that the estimation of maximum magnitude needs a revision. Since fault length is a critical parameter in their analysis the mapping procedure should be clearly explained. The authors apply most likely remote-sensing techniques but the mapping approach and the "type and quality" of base-maps is poorly given ("imagery" + 15x15 m DEM). The "morphological" criteria used to classify the faults as "active" should be given definitely. A complex definition for active faults is provided in the manuscript: "an active fault, is defined here as a plane that ground-rupturing with speeds of approximately 0.001 mm/year, with seismic activity associated, at least, in the last 10,000 years and is oriented in favour of the current stress field. The active fault planes must be related to earthquakes of magnitude Mw $\geq$ 5.4 or capable of generating rupture lengths greater than or equal to 3 km." Authors

need to show that all 316 faults fulfil that definition (for example, have all faults a minimum of 0.001 mm/yr slip-rate? Which studies provide this information? What type of information provides the CeMIEGeo database on faults?). The seismicity of the study area is concentrated to the eastern part (Figure 2). Leaving many earthquakes in the West with no earthquakes at all. How are faults satisfying the Mw $\geq$ 5.4 criteria? The entire dataset should be available for download so the results can be reproduced and tested. The text provides a theoretical but limited description of the Hurst Exponent analysis. The method tests the tendency of a time-series (here the various slip-rates given in Table 1). However, the slip-rates are controlled by the spatial distribution of the stress field and therefore have a local significance. The authors need to explain why this approach based on time-series is applicable on a dataset that has a spatial significance. In addition, more information is necessary on how slip-rates have been exactly used in the calculations. Uncertainties and error ranges are not discussed in the manuscript. What are the error ranges for the fault length and slip-rates? How do they affect the results? This questions should be addressed within the text. Similarly, the fractal analysis lacks of adequate information on the geological significance of the analysis. What is the meaning of a staircase like pattern from a tectonic/geologic perspective? In line 198 the author states the high value of the fractal dimension "may indicate the possibility" for generation of a major earthquake on the faults of the MAFS; which is a highly ambiguous result. More information is needed on how the method is applied. Which dataset is exactly used? What are the 2D boundaries of the study?

I consider the manuscript brings together an important amount of information concerning the paleoseismic results of the study area. A significant number of faults where mapped/digitzed within this study, which have a value in the seismic hazard assessment of the region. However, a distinction should be made among faults mapped within this work and obtained from other sources so readers can better evaluate the contribution of this work. In addition, the mapping approach should be defined precisely in order to evaluate the reliability of the fault map. The main results of this work are based on a statistical analysis of the fault map and paleoseismic findings. However the

results are poorly discussed and their significance in terms of active tectonics is not well addressed. I think, the applied fractal analysis is not a novel approach for evaluating the seismic hazard of a region. Authors need to improve extensively how this analysis contributes to the seismic hazard assessment. Structurally, the manuscript lacks of a proper organization. Overall, results of the work do not address sufficiently the purposes given in the text. Therefore, I consider the manuscript requires a major revision.

Further remarks on the manuscript.

1-Title: The title calls for a manuscript that actually deals with significant amount of paleoseismic field work that permit to determine new seismic sources and their characteristics. However, the work is based on mathematical approaches on previous works. I suggest to revise the title that is more compatible with the used methodology. 2-Abstract: 2/3 of the abstract is dedicated to the seismotectonics of the study area. Most of this general information is neither connected to the applied methods nor the results of this work. The abstract may get more informative if more detail is provided on the approach and methodology. Also, the significance of the results is not sufficiently and clearly expressed. 3-Figure 1 and 2 require additional information on location names, major faults systems and information on concerning the seismic activity. 4-Acambay earthquake location and related surface rupture should be given in figure 2 5-Figures 2 , add Focal Mechanisms need information for earthquake magnitude and time. 6-Slip-rates should be placed on the fault map (Figure 2) 7-Line 162: The active faults in Figure 2, do not full-fill the definition given in line 162-165. The corresponding seismic activity is not available, slip-rate estimations are missing, paleoseismic studies are missing. Therefore more information is needed on how faults are selected. 8-Figure 3 : It is unclear what it represents. Is it based on fault central points from A to B? Requires a detailed figure caption. 9-The results and discussion section contains theoretical information on the used methods, which should be placed to appropriate section. 10-Figure 4 requires more explanation. Requires labelling and a detailed figure caption.

11-In Table 1: The 2 mm/yr slip-rate for the Vento de Bravo fault could not be found in the related citation (Suter et al., 1995). 12-Table 1 Add error ranges for slip-rates, fault length and scarps. 13-Line 173-179 and 184-188: The purpose of these texts within the context of maximum magnitude is not clear.
* * *

---

## Author Comment (AC2) · 10 Aug 2018

Dear Dr Mustapha Meghraoui:

We are pleased to resubmit for publication the revised version of MS No.: nhess-2018-63 "Active Faults sources of the Morelia-Acambay Fault System, Mexico based on Paleoseismology and the estimation of magnitude Mw from fault dimensions" We appreciated your constructive criticisms.

REFEREE COMMENTS: The most substantial revision concerns the organization and

the writing of the manuscript. We have addressed each of their concerns as outlined below.

1) General remarks - The main topic of the manuscript (ms) is on the fractal fault distribution and its related seismic activity but this is not clear neither from the title, nor for the abstract and text.

**We have rewritten the title and highlighted the main objective during the text. The current title is "Active Faults Sources for the Pátzcuaro-Acambay Fault System (Mexico): Fractal Analysis of Slip Rates, and Magnitudes Mw Estimated from Fault Length".**

- This article needs to be restructured in order to clearly put forward the fractal analysis, the authors do not present new fault data and hence, the presented neotectonic and seismotectonic characteristics cannot be considered as the main topic of this article.
**We have restructured the paper to provide more clarity and highlighted the fractal analysis for the study of faults.**

-The authors mention the existence of 316 fault segments in text and about 22 fault characteristics (in Table 1) of the Morelia Acambay Graben. However, they do not explain how they did select these 22 items among the 316 faults, and which fault segments where used for the fractal analysis. The 316 fault segments deserve to be shown as a supplemental material.
**A fault database was constructed on a 15-m DEM and showed in the supplemental material. For the fractal analysis, we have used two data: (a) 316 average magnitudes Mw calculated by the surface rupture length on a 15-m Digital Elevation Model and (b) 22 slip-rates recorded in the literature. This information is reflected in section 3.Materials.**

-The seismicity and neotectonic database and related catalogs need to be clearly presented in the form of tables with appropriate legends showing the origin of data. A table of paleoseismic, historical and instrumental earthquakes is needed in this manuscript,
at least for earthquakes with Mw $\geq$ 5.4 (according to their concluding remarks).
**We have explained this information in the following sections: 2.1Paleoseismol-**
**ogy in the PAFS, and 2.2 Historical and Instrumental Seismicity in the PAFS. The**
**seismic catalog, covering from 1912 to 2018, was obtained from the Seismolog-**
**ical Service of Mexico (Servicio Sismológico Nacional, SSN; Fig. 2). The data is**
**available on their web page www.ssn.unam.mx. This catalog only has two events**
**with Mw $\geq$ 5.4 (Acambay and Maravatío earthquakes). We are showed in Fig.1 the**
**seismic events that have affected populations within the PAFS.**

-Table 1 needs to include the minimum and maximum, observed and estimated coseis-
mic slip/event for the known faults. Table 1 needs a serious legend.
**We have modified Table 1 in the revised version of the manuscript, but we have**
**decided not to include the estimated coseismic slip/event because we focus our**
**work on the temporal analysis of slip rates and on the spatial analysis of fault**
**lengths. Thus, for the temporal analysis of the details of the coseismic slip (time-**
**less term) are beyond the scope of this work. However, we have included, in the**
**subsection 2.2, maximum displacements for three faults with surface rupture**
**(Urbina and Camacho, 1913): the Acambay-Tixmadejé fault (Dmax = 50 cm), the**
**Temascalcingo fault (Dmax = 30 cm) and the Pastores fault (29 $\leq$ Dmax $\leq$ 37 cm;**
**Ortuño et al., 2015).**

- An interesting issue is the difference between the fracture density and fracture con-
centration. This section of the manuscript needs to be developed in order to show the
meaning of this difference, explain well the correlation between box dimensions and
the effects of the size of fracture concentration. The calculation of the Hurst Exponent
H and related strong persistent process, Devil staircase and box dimension should be
explained more extensively. These aspects that are fundamental in this manuscript
should appear in a separated Methodological section.
**This information is reflected in sections 4. Methods for the Study of Faults using**
**Fractal Analysis, 5. Results and 6. Discution.**

2)Specific remarks -Title: It has to be reconsidered because as presented, it shows that active faults and paleoseismic analysis are the main topic of the manuscript. I think that the fractal analysis from existing fault data should be clearly announced in this title.

**We have restructured the title according to the main topic. Traditionally this system has been named as Morelia-Acambay Fault System, in spite of, this extends to the city of Pátzcuaro. Thus, we consider that is more accurate to name it as Pátzcuaro-Acambay Fault System (PAFS).**

- Abstract: The authors use different magnitude scales (Ms, Mb, Mw). If a seismicity catalogue with homogenized magnitudes exists for Mexico, then the authors should use Mw only in this section.

**We have rewritten the Abstract, however, Ms=6.9 and Ms=5.6 was conserved because they are historical earthquakes.**

- The Introduction section is not well written, and although it includes several paragraphs as seismotectonic settings, it does not explain the geodynamic context with clear stress and strain distribution. For instance, Figures 1A and B that are redundant they show only the topography and bathymetry. Figure 1C is supposed to show the seismotectonic setting but it looks only like a geographic indication of the Morelia-Acambay Graben. The introduction needs to be better organized to explain the context and main issue, the used general methodology (fractal analysis) and its application elsewhere in comparable seismotectonic domains, previous works emphasizing the main results and finally the main steps adopted in this ms.

**We have rewritten the Introduction and remade Figure 1. The seismotectonic settings paragraphs are moved to the appropriate section.**

- (Neotectonic and seismotectonic settings?) Since the Morelia-Acambay Graben has a rich database, a specific section in neotectonics and seismotectonics would therefore be needed after the introduction. In this case, the authors should organize their text and avoid a mix of data. This section needs to present: 1) the seismicity (historical and instrumental) with emphasis on major events and their characteristics, 2) the geodetic results (GPS, conventional), focal mechanism solutions and fault kinematics for the stress and strain distribution, and 3) the paleoseismic data and results including the estimated slip rates with the corresponding time window and related uncertainties. This section has not to be long but it has to focus on major results showing the related references and how completed is the database (reference to tables in supplementary material is recommendable).
**We have divided in three subsections: 2.1 Paleoseismicity in the PAFS; 2.2 Historical and Instrumental Seismicity in the PAF; and 2.3 GPS Measurements.**

- Line 20 – 21: Please note that historical earthquakes needs to indicated with their intensities (or inferred magnitudes), their severity (number of victims whenever possible).
**We have explained this information in the subsection 2.2 Historical and Instrumental Seismicity in the PAFS.**

-Line 22: ".. set of earthquakes . . ." of what magnitudes?
**We have added the magnitudes (2.5 < Mw < 3.0) in the Introduction.**

-Line 26-27: These lines are concluding remarks and should be moved at the end of ms.
**We have moved these lines at the end of the manuscript (5.Results; 6.Discution; and 7.Conclusions).**

-Line 40: Instead of cortical, the term "crustal" is usually used in active tectonics.
**We have changed the term crustal in the Introduction.**

-Line 46: "The kinematics of them . . ." change in Their kinematics . . . This sentence mentions details on the neotectonic episodes and a reference is needed here. **We have changed the sentence in the section 2.Tectonic Setting of the PAFS.**

-Line 49: normal- right? change in Oblique fault with right-lateral normal component.
**We have changed the sentence in the section 2.Tectonic Setting of the PAFS.**

[Figure]

-Line 55: The 8.2 km depth of the Maravatio earthquake needs uncertainties. The sentence should be rewritten "Subsequently, another earthquake in 1979 with a magnitude Mb = 5.3 and a depth of 8.2 km (Astiz-Delgado, 1980), caused major damage in Maravatío."
**We have changed the sentence in the subsection 2.2Historical and Instrumental Seismicity in the PAFS.**

-Line 59: "is very probable that this sequence of earthquakes is related to the La Paloma fault of 13 km of length . . .". How did you infer this? If this is obtained from the two local stations then the "probable" should turn into "possible". Please explain. Line 59-60: ". . . active from the Holocene" does not mean much. I would suggest considered active because it affects Holocene deposits.
**It is "possible" that these earthquakes are related with the La Paloma fault because the focal mechanism is in correspondence to the fault geometry (normal fault with left-lateral component). Moreover, this fault is considered active, because it affects Holocene deposits. This information is reflected in subsection 2.2 Historical and Instrumental Seismicity in the PAFS.**

-Line 63: remove seismic risk and put seismic hazard instead.
**We have changed the term in the subsection 2.1 Paleoseismicity in the PAFS.**

-Lines 65 to 84: In all these paragraphs, slip rates need to be explained (from which field trenches and markers, e.g., lateral or vertical offset of streams, ...) and measurements span which timeframe.
**We have rewritten this paragraph in the subsection 3.3 Slip Rates and their Cumulative Distribution.**

-Line 81: What is the mechanism of the dozen faults?
**The focal mechanism corresponds to normal faulting with left-lateral components in the state of Michoacán. The three focal mechanism solutions along the PAFS reported in the literature are shown in Fig. 2. This information is reflected**

**in subsection 2.1 in the revised version of the manuscript.**

Are they in table 1?
**No, we just settled the values of the slip-rates recorded in the literature, because they are the scope of this work. The focal mechanisms in the PAFS are showed in Figure 2 in the revised manuscript and in Fig.2 in the actual response.**

-Line 90: Active faults are ...
**Revised.**

-Line 92: "... speeds of approximately ..."; fault speed is not used in active tectonics. Slip rate is more appropriate. Please apply correction throughout the text.
**Revised.**

-Line 91: The title is inappropriate in this ms. You are only extracting the data from previous works and not mapping and describing the faults of the Morelia-Acambay Graben.
**The title has been changed. In this work we have mapped 316 fault dimensions (Length and scarp) on a 15 meter Digital Elevation Model, using imagery provided by the Instituto Nacional de Estadística y Geografía (INEGI, acronym in Spanish). Additionally, we are suggesting fault names based on the names of the nearest towns, in order to homogenize nomenclature for researchers interested in correcting or completing the existing database.**

- Figure 2 is a bad quality map. Unless a clear srtm background topography can be shown, it should be removed, leaving only the seismicity and tectonic data in the map. The dates and magnitudes of focal mechanisms need to indicate in the map and in a table with their characteristics (in the supplemental material).
**The Figure 2 has been reconstructed.**

-Line 102: Unless you indicate criteria for selection, the characteristics of the 316 fault segments need to be shown at least in the supplemental material.

We have showed the 316 faults in the supplemental material.

-Lines 105 and 106: Fault length, Fault scarp height (?)
**The lengths of fault trajectories are corresponded to the lengths of mountain front sinuosity, and the scarp was measured at the maximum hillslope value for each fault.**

-Line 111: Distance between a locality and fault zone. **We have changed the statement in 3.1Mapping the Pátzcuaro-Acambay Fault System.**

-Estimation of Mw magnitudes as shown in Figures 3 a and b needs a reevaluation. Including the uncertainties of fault parameters is critical in the fractal analysis.
**The assumed error for the morphometric parameters measured was not relevant for our analysis because the lowest fault length (3000 m) is lesser than the map resolution (15 m). However, we estimate the following range 0.0002 < error <0.007 km. Figure 3 has been modified in order to show the magnitude variations from east to west (the firmagram plot). Even more, the Hurst exponent values were included for the western, central and eastern sectors of the PAFS, as well as we have printed the most known faults names.**

-The section 2.4 on the fractal analysis is devoted almost entirely to the methodological aspect; please indicate it accordingly as for instance "Method of faulting study using fractal analysis". The manuscript is mainly based on this methodology section and it should be presented before the database (seismotectonic) section.
**We have rewritten this section and added extra subsections:  4.1Self-similar Behavior in Earth Science, 4.2The Hurst Exponent, 4.3Wavelet Variance Analysis, 4.4Box Dimension, 4.5Variograms, 4.6Intensity Scale (ESI 07), and 4.7Active Fault Definition.**

-Line 149: In equation (??), please complete.
**In the last manuscript was the equation (2). In this new version corresponds to equation (1) in the subsection 4.4 Box Dimension.**

-Line 162: . . . as fault planes . . . Also remove speeds, and replace by slip rate. **We have corrected this term.**

-Line 164- 165: " . . . earthquakes of magnitude Mw $\geq$ 5.2 or related to rupture lengths greater than or equal to 3 km." Why Mw $\geq$ 5.2 and why lengths $\geq$ 3 km? How about hidden faults below Holocene deposits? As indicated by Langridge et al., (2013) and Sunye-Puchol et al., (2015) some faults can be hidden by young sedimentary deposits. In this case the fault lengths may increase. This issue needs to be discussed. **We have changed the minimum earthquake magnitude Mw $\geq$ 5.5 estimated by Wells and Coppersmith (1994), because this method is best suited for areas with crustal thickness > 15 km and avoids overestimating the magnitudes (see first paragraph of Discution). Finally, supported by Db and Hw, we can neatly determine the lower limit (3 km) of fault lengths for the PAFS. However, we cannot establish a definite upper limit due the faults hidden under Holocene deposits, not identifiable on a 15-meter Digital Elevation. We nevertheless estimated an upper limit of fault lengths (38 km) as a first approximation.**

-Line 177-178: The described seismicity, frequency and related b-value which is also a fractal distribution needs to be called earlier along with the fractal analysis in this manuscript. As this work is based on the Magana-Garcia Master thesis, that is not published and difficult to access as a reference, it should be presented with some details in introduction and seismotectonic section (or even in the supplemental material). **The seismic catalog plotted in Fig.2, covering from 1912 to 2018, was obtained from the Seismological Service of Mexico (Servicio Sismológico Nacional, SSN; Fig. 2) and the data is available on their web page: www.ssn.unam.mx. The focal mechanism parameters were reported previously by Astiz-Delgado (1980), Suter et al. (1992; 1995), Langridge et al. (2000), Singh et al. (2012), and Rodríguez-Pascua et al. (2012).**

-Line 180: Why this Table 1 is called only in section 3. This reference to the database should be called earlier!!!

We have called Table1 in subsection 3.3 Slip Rates and their Cumulative Distribution.

-Line 184: Please give a reference to the Environmental Seismic Intensity scale (ESI 07) **We have given a reference and described the Scale in section 4.6Intensity Scale (ESI 07).**

-Line 189: Hurst (1951) does not exist on the list of references.
**We have added Hurst (1951) in References section.**

-Line 191-192: The reference to the Hurst Exponent H and strong persistent process for the slip-rate distribution, along explanations on the Devil staircase should be explained in the methodology section.
**We have rewritten the methodology section and added extra subsections (4.1Self-similar Behavior in Earth Science, 4.2The Hurst Exponent, 4.3Wavelet Variance Analysis, 4.4Box Dimension, 4.5Variograms, 4.6Intensity Scale (ESI 07), and 4.7Active Fault Definition).**

-Line 192-193: "...cycles or periods with different seismic activity ...", you mean variable seismic cycles?
**This means that periodicities of earthquakes are different along the PAFS. We have rewritten the subsection 4.2The Hurst Exponent to set out the elements necessary for understanding the results of H: (a) the spatial domain, strongly suggests that the PAFS is classified in three different zones (western PAFS, central PAFS and eastern PAFS) in terms of their roughness (Hw = 0.7, Hw = 0.5, Hw = 0.8 respectively), showing different dynamics in seismotectonic activity; (b) the time-domain, with a strong persistence Hw = 0.949, suggests that the periodicities of slip rates are close in time (process with memory).**

-Line 200: This has to be included in the Methodology section.
**We have included the fracture concentration in the methodology section.**

-Line 221: What us the mathematical behaviour? You mean the mathematical or statistical expression of faulting behaviour?
**The distribution for the PAFS displays a fractal behavior, i.e. this fault population presents a self-similar behavior. This means that the log-log plot of frequency versus lengths for the PAFS obeys an inverse power law as you can see in the Fig.3 (distribution on a straight line).**

Please also note the supplement to this comment:
https://www.nat-hazards-earth-syst-sci-discuss.net/nhess-2018-63/nhess-2018-63-AC2-supplement.pdf

| EQ | DATE | MAGNITUDE | LOCATION AFFECTED | REFERENCE |
|----|------|-----------|-------------------|-----------|
| 1 | June 19th,1858 | $M_S = 7.5 - 7.7$ | Morelia and Pátzcuaro | Figueroa 1987; Garduño-Monroy et al., 1998a; García-Acosta and Suárez, 1996; Singh et al., 1996; García Acosta, 2001; Garduño-Monroy et al., 2011 |
| 2 | XIXth century | - | Zinapécuaro-Tlalpujahua | Garduño-Monroy et al., 1998b; Garduño-Monroy et al., 2009 |
| 3 | November 19th, 1912 | $M_S = 6.9$ | Acambay | Urbina and Camacho, 1913; Suter et al., 1995b, 1996 |
| 4 | February 22th,1979 | $M_s = 5.6$ | Maravatío | Astiz, 1980, 1986; Garduño-Monroy and Gutierrez-Negrín, 1990 |

**Fig. 1.** Seismic events that have affected populations within the PAFS.

| EQ | DATE | MAGNITUDE | FOCAL MECHANISM | REFERENCE |
|---|---|---|---|---|
| Acambay | November 19th, 1912 | $M_S = 6.9$ | strike=102, dip= 70, rake=-90 | Singh et al (2011); Astiz-Delgado (1980); Suter et al (1995); Suter et al (1992); Langridge et al (2000); Rodríguez- Pascua et al (2012) |
| Maravatío | February 22th,1979 | $M_s = 5.6$ | strike=280, dip= 66, rake=-48 | Astiz-Delgado(1980), Suter et al. (1992) |
| Morelia | October 17th, 2007 | $M_w = 2.7$ | strike=265, dip= 75, rake=-30 | Singh et al (2012) |

**Fig. 2.** Focal mechanism solutions in the PAFS.

- Y-axis: Log (N), ranging from 2.78 to 3
- X-axis: Log ( Length ), ranging from 3.4 to 4.6

Trend line equations shown on the plot:

$$y = -0.2593x + 3.8696$$
$$R^2 = 0.9837$$

$$y = -0.0532x + 3.0412$$
$$R^2 = 0.9086$$

[revised manuscript text omitted]

---

## Author Response (AR2)

**Dear Dr Pierre Lacan:**

We are pleased to resubmit for publication the revised version of MS No.: nhess-2018-63 "Active Faults sources of the Morelia-Acambay Fault System, Mexico based on Paleoseismology and the estimation of magnitude Mw from fault dimensions" We appreciated your constructive criticisms.

We have changed the title by request of the other referees. "Active Faults Sources for the Pátzcuaro-Acambay Fault System (Mexico): Fractal Analysis of Slip Rates, and Magnitudes Mw Estimated from Fault Length". Traditionally this system has been named as Morelia-Acambay Fault System, in spite of, this extends to the city of Pátzcuaro. Thus, we consider that is more accurate to name it as Pátzcuaro-Acambay Fault System (PAFS).

**REFERRE COMMENTS:**
The most substantial revision concerns the organization and the writing of the manuscript. We have addressed each of their concerns as outlined below.

1) General Organization:
-The structure of paper is very confused and is not easy to find the elements to follow the reasoning of the authors.
   *We have restructured the paper to provide more clarity. The sections are: 1.Introduction; 2.Tectonic Setting of the PAFS; 3.Materials; 4.Methods for the Study of Faults using Fractal Analysis; 5.Results; 6.Discution; and 7.Conclusions.*

-The introduction should introduce the problematic of the manuscript by removing all the generalities away from the objectives of the paper.
   *We have restructured the Introduction and we have highlighted the problem in the study area (Page 2, line 16-31 and Page 3, line 1-16).*

-The Seismotectonic Setting should be organized to set out the elements necessary for understanding the discussion. In its current form, everything is underneath and the state of art is not clear.
   *We have restructured the Seismotectonic Setting and added the following subsections: 2.1Paleoseismicity in the PAFS; 2.2Historical and Instrumental Seismicity in the PAFS and 2.3GPS measurements (Page 3, 4, 5, 6 and 7).*

- What morphological evidences have been taken into account delimiting fault segments and main faults. Why so much difference with works published previously. In particular Lacan et al., 2018 calculated 48 km length for the Venta de Bravo fault, you should explain how do you calculate the different length (32.982 km?) and why is this difference so important. Same for the Pastores Fault: 33 km for Langridge et al., 2013 and 38 km for you? and other faults.
   *We identified and defined fault segments on a 15-meter Digital Elevation Model. We used the imagery provided by the Instituto Nacional de Estadística y Geografía*

*(INEGI, acronym in Spanish). The criterion for the tracing of fault segments was the union of small traces to form a larger one, but only if the geomorphological continuity was clear. The lengths of fault trajectories, which is the main object of study, corresponded to the lengths of mountain front sinuosity, and the scarp was measured at the maximum hillslope value for each fault.*

*We have expected differences in length with both the previous and the most recent works, due to the different resolutions and techniques used in each study. In cases such as Venta de Bravo fault, where there are several segments that may be the continuation of the same fault, but we do not know exactly which of them are the correct ones based on the 15-meter DEM, are managed as separate segments. However, we are open to improving the fault traces with better resolutions in future works. This information is reflected in subsection 3.1Mapping the Pátzcuaro-Acambay Fault System (Page 7, line 23-30).*

-The methods should be explained more carefully.
*We have rewritten the Methods section and we changed the name section for Methods for the Study of Faults using Fractal Analysis (Page 9, line 12).*

- For the "2.4 Fractal analysis", you lose the reader with details explanations, but you do not explain what you want to calculate? Why do you think it's fractal? What does these calculations represent?
*We have rewritten this section and added extra subsections (4.1Self-similar Behavior in Earth Science, 4.2The Hurst Exponent, 4.3Wavelet Variance Analysis, 4.4Box Dimension, 4.5Variograms, 4.6Intensity Scale (ESI 07), and 4.7Active Fault Definition) to provide more clarity (Page 9-13). Here we can manage the fractal analysis because this fault population presents a self-similar behavior. This means that the log-log plot of frequency versus lengths for the PAFS obeys an inverse power law as you can see in Figure1 (distribution on a straight line). Discontinuous red lines represent the linear regression model fitted using the least squares approach. In the Results section we have mentioned that this power law is binomial, because present two slopes values (Page 13, line 15-18). This bimodality may reveal the existence of at least two different fracture processes in the PAFS. For more detail, we decided to use the Hurst analysis to delineate different zones of deformation processes. Finally, we have characterized by quantitative parameters the dynamics of the seismotectonic activity along the PAFS as we discuss during the Results and Discussion (Page 13-16).*

[Figure]

***Figure 1. Log-log plot of frequency versus lengths for the PAFS obeys an inverse power law***

- The Result and Discussion part is confused. I strongly recommend separating the results from one side (explaining the results you get) and after, a discussion section where you discuss these results and their consequences. In the current form, we do not distinguish what is new from what was already known.

   ***We have separated the Results and Discussion (Page 13-16)..***

-In particular I do not understand the relationship between the results you present and the generation of major earthquakes. What is already known, including previous mapping of faulting in the area should be carefully presented in the seismotectonic setting.

   ***The research involves fault lengths and its corresponding magnitudes Mw (spatial analysis) and slip-rates estimations of earlier studies (time analysis). The fractal method using in both the spatial and time domains allow to distinguish a non-random system and to identify the persistence of a trend within a time series (here slip-rates by the Hurst Exponent) and the micro-regionalization for the PAFS (spatial analysis of Mw by the Hurst Exponent).***

- The conclusion is also confused. You should clearly state

   ***We have restructured the section to provide more clarity. According to our analysis, we conclude that (1) the expected mean maximum earthquake magnitude for the study area was Mw 7.0, (2) we defined a micro-regionalization for the PAFS (western, central and eastern) zones by the Hurst exponent based on magnitudes Mw and (3) we have validated the intrinsic definition of active fault proposed here by fractal analysis and variograms analysis (Page 16, line 20-32 and page 17, line 1-15).***

**Dear Dr Mustapha Meghraoui:**

We are pleased to resubmit for publication the revised version of MS No.: nhess-2018-63 "Active Faults sources of the Morelia-Acambay Fault System, Mexico based on Paleoseismology and the estimation of magnitude Mw from fault dimensions" We appreciated your constructive criticisms.

**REFERRE COMMENTS:**
The most substantial revision concerns the organization and the writing of the manuscript. We have addressed each of their concerns as outlined below.

**1) General remarks**
- The main topic of the manuscript (ms) is on the fractal fault distribution and its related seismic activity but this is not clear neither from the title, nor for the abstract and text.
*We have rewritten the title and highlighted the main objective during the text. We have changed the title as "Active Faults Sources for the Pátzcuaro-Acambay Fault System (Mexico): Fractal Analysis of Slip Rates, and Magnitudes Mw Estimated from Fault Length". Traditionally this system has been named as Morelia-Acambay Fault System, in spite of, this extends to the city of Pátzcuaro. Thus, we consider that is more accurate to name it as Pátzcuaro-Acambay Fault System (PAFS).*

- This article needs to be restructured in order to clearly put forward the fractal analysis, the authors do not present new fault data and hence, the presented neotectonic and seismotectonic characteristics cannot be considered as the main topic of this article.
*We have restructured the paper to provide more clarity and highlighted the fractal analysis for the study of faults.*

-The authors mention the existence of 316 fault segments in text and about 22 fault characteristics (in Table 1) of the Morelia Acambay Graben. However, they do not explain how they did select these 22 items among the 316 faults, and which fault segments where used for the fractal analysis. The 316 fault segments deserve to be shown as a supplemental material.
*A fault database was constructed on a 15-m DEM and is showed in the supplemental material. For the fractal analysis, we have used two data: (a) 316 average magnitudes Mw calculated by the surface rupture length on a 15-m Digital Elevation Model and (b) 22 slip-rates recorded in the literature. This information is reflected in section 3.Materials (Page 7, line 22-30, page 8 and page 9, line 4-11).*

-The seismicity and neotectonic database and related catalogs need to be clearly presented in the form of tables with appropriate legends showing the origin of data. A table of paleoseismic, historical and instrumental earthquakes is needed in this manuscript, at least for earthquakes with Mw ≥ 5.4 (according to their concluding remarks).
*We have explained this information in the following sections: 2.1Paleoseismology in the PAFS  (Page 5, line 16-34 and page 6, line 1-14) and 2.2 Historical and Instrumental*

*Seismicity in the PAFS (Page 6, line 21-30 and page 7, line 1-14). The seismic catalog, covering from 1912 to 2018, was obtained from the Seismological Service of Mexico (Servicio Sismológico Nacional, SSN; Fig. 2). The data is available on their web page www.ssn.unam.mx. This catalog only has two events with Mw ≥ 5.4 (Acambay and Maravatío earthquakes). We are showed in Table 1 the seismic events that have affected populations within the PAFS.*

-Table 1 needs to include the minimum and maximum, observed and estimated coseismic slip/event for the known faults. Table 1 needs a serious legend.

*We have modified Table 1 in page 29, but we have decided not to include the estimated coseismic slip/event because we focus our work on the temporal analysis of slip rates and on the spatial analysis of fault lengths. Thus, for the temporal analysis of the details of the coseismic slip (timeless term) are beyond the scope of this work. However, we have included, in the subsection 2.2 Historical and Instrumental Seismicity in the PAFS, maximum displacements for three faults with surface rupture (Urbina and Camacho, 1913): the Acambay-Tixmadejé fault (Dmax = 50 cm), the Temascalcingo fault (Dmax = 30 cm) and the Pastores fault (29 ≤ Dmax ≤ 37 cm; Ortuño et al., 2015). Page 6, line 21-26.*

- An interesting issue is the difference between the fracture density and fracture concentration. This section of the manuscript needs to be developed in order to show the meaning of this difference, explain well the correlation between box dimensions and the effects of the size of fracture concentration. The calculation of the Hurst Exponent H and related strong persistent process, Devil staircase and box dimension should be explained more extensively. These aspects that are fundamental in this manuscript should appear in a separated Methodological section.

*This information is reflected in sections 4. Methods for the Study of Faults using Fractal Analysis, 5. Results and 6. Discussion.*

**2)Specific remarks**

-Title: It has to be reconsidered because as presented, it shows that active faults and paleoseismic analysis are the main topic of the manuscript. I think that the fractal analysis from existing fault data should be clearly announced in this title.

*We have restructured the title according to the main topic. Traditionally this system has been named as Morelia-Acambay Fault System, in spite of, this extends to the city of Pátzcuaro. Thus, we consider that is more accurate to name it as Pátzcuaro-Acambay Fault System (PAFS).*

- Abstract: The authors use different magnitude scales (Ms, Mb, Mw). If a seismicity catalogue with homogenized magnitudes exists for Mexico, then the authors should use Mw only in this section.

*We have rewritten the Abstract, however, Ms=6.9 and Ms=5.6 was conserved because they are historical earthquakes (Page 1, line 7-8).*

- The Introduction section is not well written, and although it includes several paragraphs as seismotectonic settings, it does not explain the geodynamic context with clear stress and strain distribution. For instance, Figures 1A and B that are redundant they show only the topography and bathymetry. Figure 1C is supposed to show the seismotectonic setting but it looks only like a geographic indication of the Morelia-Acambay Graben. The introduction needs to be better organized to explain the context and main issue, the used general methodology (fractal analysis) and its application elsewhere in comparable seismotectonic domains, previous works emphasizing the main results and finally the main steps adopted in this ms.

*We have rewritten the Introduction and remade Figure 1 (Page 24). The seismotectonic settings paragraphs are moved to the appropriate section (Page 6, line 21-30 and page 7, line 1-14).*

- (Neotectonic and seismotectonic settings?) Since the Morelia-Acambay Graben has a rich database, a specific section in neotectonics and seismotectonics would therefore be needed after the introduction. In this case, the authors should organize their text and avoid a mix of data. This section needs to present: 1) the seismicity (historical and instrumental) with emphasis on major events and their characteristics, 2) the geodetic results (GPS, conventional), focal mechanism solutions and fault kinematics for the stress and strain distribution, and 3) the paleoseismic data and results including the estimated slip rates with the corresponding time window and related uncertainties. This section has not to be long but it has to focus on major results showing the related references and how completed is the database (reference to tables in supplementary material is recommendable).

*We have divided in three subsections: 2.1 Paleoseismicity in the PAFS; 2.2 Historical and Instrumental Seismicity in the PAF; and 2.3 GPS Measurements (Page 3-7).*

- Line 20 – 21: Please note that historical earthquakes needs to indicated with their intensities (or inferred magnitudes), their severity (number of victims whenever possible).

*We have explained this information in the subsection 2.2 Historical and Instrumental Seismicity in the PAFS (Page 6, line 21-30 and page 7, line 1-14).*

-Line 22: ".. set of earthquakes . . ." of what magnitudes?

*We have added the magnitudes (2.5 < Mw < 3.0) in the Introduction (Page 2, line 23).*

-Line 26_27: These lines are concluding remarks and should be moved at the end of ms.

*We have moved these lines at the end of the manuscript (5.Results; 6.Discussion; and 7.Conclusions).*

-Line 40: Instead of cortical, the term "crustal" is usually used in active tectonics.

*We have changed the term crustal in the Introduction (Page 2, line 20).*

-Line 46: "The kinematics of them . . ." change in Their kinematics . . . This sentence mentions details on the neotectonic episodes and a reference is needed here.

*We have changed the sentence in the section 2.Tectonic Setting of the PAFS (Page 4, line 8).*

-Line 49: normal- right? change in Oblique fault with right-lateral normal component.

*We have changed the sentence in the section 2.Tectonic Setting of the PAFS (Page 4, line 12).*

-Line 55: The 8.2 km depth of the Maravatio earthquake needs uncertainties. The sentence should be rewritten "Subsequently, another earthquake in 1979 with a magnitude Mb = 5.3 and a depth of 8.2 km (Astiz-Delgado, 1980), caused major damage in Maravatío."

*We have changed the sentence in the subsection 2.2Historical and Instrumental Seismicity in the PAFS (Page 6, line 30).*

-Line 59: "is very probable that this sequence of earthquakes is related to the La Paloma fault of 13 km of length . . .". How did you infer this? If this is obtained from the two local stations then the "probable" should turn into "possible". Please explain. Line 59-60: ". . . active from the Holocene" does not mean much. I would suggest considered active because it affects Holocene deposits.

*It is very likely that these earthquakes are related with the La Paloma fault because the focal mechanism is in correspondence to the fault geometry (normal fault with left-lateral component). Moreover, this fault is considered active, because it affects Holocene deposits. This information is reflected in subsection 2.2 Historical and Instrumental Seismicity in the PAFS (Page 7, line 8).*

-Line 63: remove seismic risk and put seismic hazard instead.

*We have changed the term in the subsection 2.1 Paleoseismicity in the PAFS (Page 5, line 18).*

-Lines 65 to 84: In all these paragraphs, slip rates need to be explained (from which field trenches and markers, e.g., lateral or vertical offset of streams, ...) and measurements span which timeframe.

*We have rewritten this paragraph in the subsection 3.3 Slip Rates and their Cumulative Distribution (Page 8. Line 23-30).*

-Line 81: What is the mechanism of the dozen faults?

*The focal mechanism corresponds to normal faulting with left-lateral components in the state of Michoacán. The three focal mechanism solutions along the PAFS reported in the literature are shown in Fig. 2 (Page 25).*

Are they in table 1?

*No, we just settled the values of the slip-rates recorded in the literature, because they are the scope of this work. The focal mechanisms in the PAFS are showed in Figure 2 in the revised manuscript and in Table 2 in the actual response.*

-Line 90: Active faults are ...
*Revised (Page 13, line 2).*

-Line 92: "... speeds of approximately ..."; fault speed is not used in active tectonics. Slip rate is more appropriate. Please apply correction throughout the text.
*Revised (Page 13, line 3).*

-Line 91: The title is inappropriate in this ms. You are only extracting the data from previous works and not mapping and describing the faults of the Morelia-Acambay Graben.
*The title has been changed, but in this work, we have mapped 316 fault dimensions (Length and scarp) on a 15-meter Digital Elevation Model, using imagery provided by the Instituto Nacional de Estadística y Geografía (INEGI, acronym in Spanish). Additionally, we are suggesting fault names based on the names of the nearest towns, in order to homogenize nomenclature for researchers interested in correcting or completing the existing database.*

- Figure 2 is a bad quality map. Unless a clear srtm background topography can be shown, it should be removed, leaving only the seismicity and tectonic data in the map. The dates and magnitudes of focal mechanisms need to indicate in the map and in a table with their characteristics (in the supplemental material).
*Figure 2 has been reconstructed.*

-Line 102: Unless you indicate criteria for selection, the characteristics of the 316 fault segments need to be shown at least in the supplemental material.
*We have showed the 316 faults in the supplemental material.*

-Lines 105 and 106: Fault length, Fault scarp height (?)
*The lengths of fault trajectories are corresponded to the lengths of mountain front sinuosity, and the scarp was measured at the maximum hillslope value for each fault (Page 7, line24-30).*

-Line 111: Distance between a locality and fault zone.
*We have changed the statement in 3.1Mapping the Pátzcuaro-Acambay Fault System (Page 8, line 11).*

-Estimation of Mw magnitudes as shown in Figures 3 a and b needs a reevaluation. Including the uncertainties of fault parameters is critical in the fractal analysis.
*The assumed error for the morphometric parameters measured was not relevant for our analysis because the lowest fault length (3000 m) is lesser than the map resolution (15 m, Page 8, line 3-5). However, we estimate the following range 0.0002 < error <0.007 km. Figure 3 has been modified in order to show the magnitude variations from east to*

*west (the firmagram plot). Even more, the Hurst exponent values were included for the western, central and eastern sectors of the PAFS, as well as we have printed the most known faults names.*

-The section 2.4 on the fractal analysis is devoted almost entirely to the methodological aspect; please indicate it accordingly as for instance "Method of faulting study using fractal analysis". The manuscript is mainly based on this methodology section and it should be presented before the database (seismotectonic) section.

*We have rewritten this section and added extra subsections: 4.1Self-similar Behavior in Earth Science, 4.2The Hurst Exponent, 4.3Wavelet Variance Analysis, 4.4Box Dimension, 4.5Variograms, 4.6Intensity Scale (ESI 07), and 4.7Active Fault Definition (Page 9-13).*

-Line 149: In equation (??), please complete.

*In the last manuscript was the equation (2). In this new version corresponds to equation (1) in the subsection 4.4 Box Dimension (Page 11, line 27).*

-Line 162: . . . as fault planes . . . Also remove speeds, and replace by slip rate.

*We have corrected this term (Page 13, line 2-3).*

-Line 164- 165: " . . . earthquakes of magnitude Mw ≥ 5.2 or related to rupture lengths greater than or equal to 3 km." Why Mw ≥ 5.2 and why lengths ≥ 3 km? How about hidden faults below Holocene deposits? As indicated by Langridge et al., (2013) and Sunye-Puchol et al., (2015) some faults can be hidden by young sedimentary deposits. In this case the fault lengths may increase. This issue needs to be discussed.

*We have changed the minimum earthquake magnitude Mw ≥ 5.5 estimated by Wells and Coppersmith (1994) relation, because this method is best suited for areas with crustal thickness > 15 km and avoids overestimating the magnitudes (see first paragraph of Discussion, page 14, line 5-14). Finally, supported by $D_b$ and $H_w$, we can neatly determine the lower limit (3 km) of fault lengths for the PAFS. However, we cannot establish a definite upper limit due the faults hidden under Holocene deposits, not identifiable on a 15-meter Digital Elevation. We nevertheless estimated an upper limit of fault lengths (38 km) as a first approximation.*

-Line 177-178: The described seismicity, frequency and related b-value which is also a fractal distribution needs to be called earlier along with the fractal analysis in this manuscript. As this work is based on the Magana-Garcia Master thesis, that is not published and difficult to access as a reference, it should be presented with some details in introduction and seismotectonic section (or even in the supplemental material).

*The seismic catalog plotted in Fig.2 (page 25), covering from 1912 to 2018, was obtained from the Seismological Service of Mexico (Servicio Sismológico Nacional, SSN; Fig. 2). The data is available on their web page: www.ssn.unam.mx. The focal mechanism parameters were reported previously by Astiz-Delgado (1980), Suter et al. (1992; 1995), Langridge et al. (2000), Singh et al. (2012), and Rodríguez-Pascua et al. (2012).*

-Line 180: Why this Table 1 is called only in section 3. This reference to the database should be called earlier!!!

*We have called Table1 in subsection 3.3 Slip Rates and their Cumulative Distribution (Page 8, line 30).*

-Line 184: Please give a reference to the Environmental Seismic Intensity scale (ESI 07)

*We have given a reference and described the Scale in section 4.6Intensity Scale (ESI 07). Page 12, line 25-30.*

-Line 189: Hurst (1951) does not exist on the list of references.

*We have added Hurst (1951) in References section (Page 19, line 25-26).*

-Line 191-192: The reference to the Hurst Exponent H and strong persistent process for the slip-rate distribution, along explanations on the Devil staircase should be explained in the methodology section.

*We have rewritten the methodology section and added extra subsections (4.1Self-similar Behavior in Earth Science, 4.2The Hurst Exponent, 4.3Wavelet Variance Analysis, 4.4Box Dimension, 4.5Variograms, 4.6Intensity Scale (ESI 07), and 4.7Active Fault Definition). Page 9-13.*

-Line 192-193: "...cycles or periods with different seismic activity ...", you mean variable seismic cycles?

*This means that periodicities of earthquakes are different along the PAFS. We have rewritten the subsection 4.2The Hurst Exponent (Page 10, line 10-30 and page 11, line 1-13) to set out the elements necessary for understanding the results of H: (a) the spatial domain, strongly suggests that the PAFS is classified in three different zones (western PAFS, central PAFS and eastern PAFS) in terms of their roughness (Hw = 0.7, Hw = 0.5, Hw = 0.8 respectively), showing different dynamics in seismotectonic activity; (b) the time-domain, with a strong persistence Hw = 0.949, suggests that the periodicities of slip rates are close in time (process with memory).*

-Line 200: This has to be included in the Methodology section.

*We have included the fracture concentration in the methodology section (Page 9, line27-31 ang page 10, line 1).*

-Line 221: What us the mathematical behaviour? You mean the mathematical or statistical expression of faulting behaviour?

*The distribution for the PAFS displays a fractal behavior, i.e. this fault population presents a self-similar behavior. This means that the log-log plot of frequency versus lengths for the PAFS obeys an inverse power law as you can see in the Fig.3 (distribution on a straight line) in the actual response.*

| EQ | DATE | MAGNITUDE | LOCATION AFFECTED | REFERENCE |
|----|------|-----------|-------------------|-----------|
| 1 | June 19th,1858 | $M_S$ = 7.5 – 7.7 | Morelia and Pátzcuaro | Figueroa 1987; Garduño-Monroy et al., 1998a; García-Acosta and Suárez, 1996; Singh et al., 1996; García Acosta, 2001; Garduño-Monroy et al., 2011 |
| 2 | XIXth century | - | Zinapécuaro-Tlalpujahua | Garduño-Monroy et al., 1998b; Garduño-Monroy et al., 2009 |
| 3 | November 19th, 1912 | $M_S$ = 6.9 | Acambay | Urbina and Camacho, 1913; Suter et al., 1995b, 1996 |
| 4 | February 22th,1979 | $M_s$ = 5.6 | Maravatío | Astiz, 1980, 1986; Garduño-Monroy and Gutierrez-Negrín, 1990 |

*Table 1 Seismic events that have affected populations within the PAFS.*

| EQ | DATE | MAGNITUDE | FOCAL MECHANISM | REFERENCE |
|----|------|-----------|-----------------|-----------|
| Acambay | November 19th, 1912 | $M_S$ = 6.9 | strike=102, dip= 70, rake=-90 | Singh et al (2011); Astiz-Delgado (1980); Suter et al (1995); Suter et al (1992); Langridge et al (2000); Rodríguez- Pascua et al (2012) |
| Maravatío | February 22th,1979 | $M_s$ = 5.6 | strike=280, dip= 66, rake=-48 | Astiz-Delgado(1980), Suter et al. (1992) |
| Morelia | October 17th, 2007 | $M_w$ = 2.7 | strike=265, dip= 75, rake=-30 | Singh et al (2012) |

*Table 2: Focal mechanism solutions in the PAFS.*

[Figure]

***Fig. 3 Log-log plot of frequency versus lengths for the PAFS obeys an inverse power law***

**Dear Professor Aksoy:**

We are pleased to resubmit for publication the revised version of MS No.: nhess-2018-63 "Active Faults sources of the Morelia-Acambay Fault System, Mexico based on Paleoseismology and the estimation of magnitude Mw from fault dimensions" We appreciated your constructive criticisms.

**REFEREE COMMENTS:**
The most substantial revision concerns with the need of significant improvements on the presentation and structure of the work; more information methodology, the approach and the significance of the results. We have addressed each of your concerns as outlined below.

1) General Organization:
-The figures lack significantly of useful information that are necessary to comprehend the study area. Many cities, locations, fault names mentioned in the text are not available in maps and figures, making it difficult for the reader to orient him/herself spatially.
> ***We have restructured Figure 1, 2 and 3 (Page 24-26).***

- Although the authors provide some theoretical information on the statistical calculations the connection and relation to the seismic hazard evaluation is poorly given, the geological significance for each input and output are not provided and discussed in the manuscript sufficiently.
> ***We have restructured the paper to provide more clarity and highlighted the fractal analysis for the study of faults.***

-The aim of the study is confusing because throughout the text authors describe several different purposes: 1-prepare an intrinsic definition for active faults (abstract) 2-estimation of possible maximum earthquake magnitudes (abstract) 3-understand the seismic activity from Patzcuaro to Acambay sector (introduction) 4-define the intracontinental structures that are susceptible to generate moderate and strong seismic events (line 85).

Aside the quantitative results, the study addresses only the first two purposes clearly. Maximum earthquake magnitudes are calculated via fault length measurements and a comprehensive definition is given for active faults. Based on the Hurst Exponent it is concluded that the fault system is active, however the possibility of an inactive fault system is not discussed within the manuscript.

*We have rewritten the aims of the study. The goals have been mentioned in the introduction (page 3, line 12-16) and they are: (1) the estimation of the maximum possible earthquake magnitudes by three empirical relations; (2) the definition of a micro-regionalization of the PAFS using the Hurst exponent based on Mw magnitudes; and (3) the validation of our proposed definition of Active Fault sources for the PAFS by fractal analysis and semivariograms. Consequently, we are proposing the investigation of the dynamics of the Pátzcuaro-Acambay area, in order to improve territory planning and reduce seismic hazard.*

-The authors illustrate (Fig 3) that these relationships give different magnitude estimations for the same fault section but do not discuss how they interpret this difference. No reasoning is provided why authors prefer to take into account the Wesnousky (2008) relationship.

*The analysis of the three empirical relations results of active faults was summarized as follows: (1) The model proposed by Anderson et al. (1996) always yields lower results than the other relationships; however, (2) the highest magnitudes are obtained with the relationship of Wesnousky (2008); (3) the average magnitudes are obtained by means of Wells and Coppersmith (1994) relationship. We have observed that all three relationships work for the PAFS. However, in this paper, we reported the maximum and minimum earthquake magnitudes estimated by Wells and Coppersmith (1994), because this method is best suited for areas with crustal thickness > 15 km and avoids overestimating the magnitudes (see Fig.3 in page 26).*

-The analysis assumes that each fault section has the potential to rupture the entire crust individually; (at all scales like 3-5 km). Why is 3 km the minimum preferred fault length that is included into the dataset? Can these faults also create surface ruptures?

*A key step in this study was to delimit the minimum fault length. For this purpose, we made a test to find the fractal capacity dimension $D_b$ for our database which contains 316 average Mw magnitudes calculated from the surface rupture length by the use of three different empiric relationships. We also calculate the $D_b$ for the same database but including faults less than 3 km (a total of 628 faults). The results were: $D_b$ (316)= 1.33, $D_b$ (628)= 1.77*

*Based on the results of Nieto-Samaniego et al. (2005), they proved that box dimension is in inverse relation with fracture concentration. Moreover, Poulimenos (2000), Cowie et al. (1995), Ackermann et al. (2001) have also shown that the total fracture length is directly proportional to the amount of deformation, i.e., large fractures can accommodate more deformation than small ones. Consequently, we have inferred that the low value of the fractal dimension Db (316)= 1.33 corresponds to greater amounts of large fault lengths: it is well-known that large fractures can accommodate more deformation. Thus, we are interested in the minimum earthquake magnitudes Mw ≥ 5.5 (or SRL=3km) for improving the vulnerability studies, because is acutely necessary in the central portion of Mexico.*

-Furthermore, this analysis needs to consider the spatial distribution and interaction of the faults. An earthquake may rupture several adjacent fault segments; which would necessarily imply a larger earthquake magnitude. Authors need to consider multi-segment ruptures according to fault segmentation patterns and spatial distribution of the faults. Therefore, I consider that the estimation of maximum magnitude needs a revision. Since fault length is a critical parameter in their analysis the mapping procedure should be clearly explained. The authors apply most likely remote-sensing techniques but the mapping approach and the "type and quality" of base-maps is poorly given ("imagery" + 15x15 m DEM). The "morphological" criteria used to classify the faults as "active" should be given definitely.

*Notwithstanding the importance of this kind of study, the linkage mechanism is beyond the scope of this work, due that we need to know the maximum possible length for each fault, and for this purpose we need a DEM with more resolution. Based on the Db and Hw results, we can neatly determine the lower limit (3 km) of fault lengths for the PAFS, but we cannot establish a definitive upper limit due the faults hidden under Holocene deposits, not identifiable on a 15-meter Digital Elevation. We nevertheless estimated an upper limit of fault lengths (38 km) as a first approximation (Page 16, line 17-19).*

-A complex definition for active faults is provided in the manuscript: "an active fault, is defined here as a plane that ground-rupturing with speeds of approximately 0.001 mm/year, with seismic activity associated, at least, in the last 10,000 years and is oriented in favour of the current stress field. The active fault planes must be related to earthquakes of magnitude Mw ≥ 5.4 or capable of generating rupture lengths greater than or equal to 3 km." Authors need to show that all 316 faults fulfil that definition (for example, have all faults a minimum of 0.001 mm/yr slip-rate? Which studies provide this information?

*This is the first study that works with a set of slip rate estimations in the system with a fractal approach. We support our results based on the fractal analysis for this set: Figure 4 (Page 27) shows that Db = 1.86 is related to a lower concentration of low slip rates in the PAFS, suggesting that larger faults accommodate the strain more efficiently; and with a strong persistence (Hw = 0.949), i.e. the periodicities of slip rates are close in time (process with memory). Moreover, active faults are optimally oriented to the current stress field, in terms of variogram analysis, an anisotropic direction was identified in ENE direction (80º, Fig.5, page 28), as well as the active fault planes are related to earthquakes with a minimum magnitude of Mw ≥ 5.5, or capable of generating rupture lengths greater than*

*or equal to 3 km. Thus, we can prove, in terms of fractal analysis, that the 316 faults studied for the PAFS are seismically active.*

- What type of information provides the CeMIEGeo database on faults?

*The CeMIEGeo provides fault length information around the Cuitzeo Lake (Page 3, line 30).*

-The seismicity of the study area is concentrated to the eastern part (Figure 2). Leaving many earthquakes in the West with no earthquakes at all. How are faults satisfying the Mw ≥ 5.4 criteria?

*We have restructured the Seismotectonic Setting and added the following subsections: 2.1Paleoseismicity in the PAFS; 2.2Historical and Instrumental Seismicity in the PAFS and 2.3GPS measurements. Here, we set out that The Pátzcuaro-Acambay Fault System can be divided into three zones with different geological and geophysical settings (page 4, line 14-35 and page 5, line 1-15), and in the Results and Discussion we present the Hurst analysis. The results in the spatial domain strongly suggest that the PAFS is classified in three different zones (western PAFS, central PAFS and eastern PAFS) in terms of their roughness (Hw = 0.7, Hw = 0.5, and Hw = 0.8 respectively; Fig. 3), with their corresponding magnitudes (5.5 ≤ Mw ≤ 6.9; 5.5 ≤ Mw ≤ 6.7; 5.5 ≤ Mw ≤ 7.0)(Page 13, line 19-21). As we can see in the historical seismicity, paleoseismology studies and the spatial distribution of faults in the western zone, the faults are capable to generate earthquakes with magnitudes 5.5 ≤ Mw ≤ 6.9. Thus, we strongly believe that the area must continue to be monitoring in order to reduce seismic hazard in central Mexico.*

-The entire dataset should be available for download so the results can be reproduced and tested.

*The datasets generated during the current study are available from the corresponding author on reasonable request. The seismic catalog, covering from 1912 to 2018, was obtained from the Seismological Service of Mexico (Servicio Sismológico Nacional, SSN) and it is available on the web page www.ssn.unam.mx.*

-The text provides a theoretical but limited description of the Hurst Exponent analysis. The method tests the tendency of a time-series (here the various slip-rates given in Table 1). However, the slip-rates are controlled by the spatial distribution of the stress field and therefore have a local significance. The authors need to explain why this approach based on time-series is applicable on a dataset that has a spatial significance. In addition, more information is necessary on how slip-rates have been exactly used in the calculations.

*We have rewritten this section and added extra subsections for more clarity: 4.1Self-similar Behavior in Earth Science, 4.2The Hurst Exponent, 4.3Wavelet Variance Analysis, 4.4Box Dimension, 4.5Variograms, 4.6Intensity Scale (ESI 07), and 4.7Active Fault Definition (Page 9-13).*

-Uncertainties and error ranges are not discussed in the manuscript. What are the error ranges for the fault length and slip-rates? How do they affect the results? This questions should be addressed within the text.

*The assumed error for the morphometric parameters measured was not relevant for our analysis because the lowest fault length (3000 m) is lesser than the map resolution (15 m). However, we estimate the following range 0.0002 < error <0.007 km. This information is reflected in subsection 3.1Mapping the Pátzcuaro-Acambay Fault System in page 8, line 3-6. As we can see in table 1, not all the slip-rates errors are reported by previous authors, and they are in the range of 0.02mm/yr.*

-Similarly, the fractal analysis lacks of adequate information on the geological significance of the analysis. What is the meaning of a staircase like pattern from a tectonic/geologic perspective?

*There is a dependence and causality between the Hurst Exponents, fractal Dimension and the PAFS dynamics, we have developed this topic in the Results and Discussion section in the page 13-16.*

-In line 198 the author states the high value of the fractal dimension "may indicate the possibility" for generation of a major earthquake on the faults of the MAFS; which is a highly ambiguous result. More information is needed on how the method is applied. Which dataset is exactly used? What are the 2D boundaries of the study?

*Spatial-temporal methods were applied to the active fault data, and the fractal behavior observed for the entire PAFS allows us to define that the PAFS is seismically active. This is supported by the results of the Hurst analysis for the fault lengths and its corresponding magnitudes Mw (spatial analysis) as well as the slip-rates estimations of earlier studies (time analysis). We have included a better explanation in the section 5. Results in the page 13, line 11-32.*

-However, a distinction should be made among faults mapped within this work and obtained from other sources so readers can better evaluate the contribution of this work.

*This is the first study that works with a fault population in the PAFS defining a total of 316 active faults with fault dimensions (Length and scarp) on a 15-meter Digital Elevation Model with a fractal approach. Moreover, we estimated 316 average Mw magnitudes calculated from the surface rupture length by the use of three different empiric relationships. We perform a major revision in the revised version of the manuscript in order to explain the contribution of our work along the entire document.*

-In addition, the mapping approach should be defined precisely in order to evaluate the reliability of the fault map.

*The criterion for the tracing of fault segments was the union of small traces to form a larger one, but only if the geomorphological continuity was clear. The lengths of fault trajectories, which is the main object of study, corresponded to the lengths of mountain front sinuosity, and the scarp was measured at the maximum hillslope value for each fault (page 7, line 26-30).*

*We have expected differences in length with both the previous and the most recent works, due to the different resolutions and techniques used in each study. However, we are open to improving the fault traces with better resolutions in future works. This information is reflected in subsection 3.1Mapping the Pátzcuaro-Acambay Fault System.*

-The main results of this work are based on a statistical analysis of the fault map and paleoseismic findings. However the results are poorly discussed and their significance in terms of active tectonics is not well addressed.

*We have rewritten the Results, Discussion and Conclusion for clarify how this analysis contributes to the seismic hazard assessment (Page 13-17).*

2) Further remarks on the manuscript

1-Title: The title calls for a manuscript that actually deals with significant amount of paleoseismic field work that permit to determine new seismic sources and their characteristics. However, the work is based on mathematical approaches on previous works. I suggest to revise the title that is more compatible with the used methodology.

*We have changed the title and highlighted the main objective during the text. The current title is "Active Faults Sources for the Pátzcuaro-Acambay Fault System (Mexico): Fractal Analysis of Slip Rates, and Magnitudes Mw Estimated from Fault Length". Traditionally this system has been named as Morelia-Acambay Fault System, in spite of, this extends to the city of Pátzcuaro. Thus, we consider that is more accurate to name it as Pátzcuaro-Acambay Fault System (PAFS).*

2- Abstract: 2/3 of the abstract is dedicated to the seismotectonics of the study area. Most of this general information is neither connected to the applied methods nor the results of this work. The abstract may get more informative if more detail is provided on the approach and methodology. Also, the significance of the results is not sufficiently and clearly expressed.

*We have restructured the abstract to provide more clarity (Page 1 and page 2, line 1-14).*

3-Figure 1 and 2 require additional information on location names, major faults systems and information on concerning the seismic activity. 4-Acambay earthquake location and related surface rupture should be given. 5-Add Focal Mechanisms need information for earthquake magnitude and time. 6-Slip-rates should be placed on the fault map. 7- The corresponding seismic activity is not available.

*Figure 1 and 2 have been reconstructed (Page 24-25).*

8-Figure 3: It is unclear what it represents. Is it based on fault central points from A to B? Requires a detailed figure caption.

*The profile A-B was about the depth of seismicity and we removed the profile A-B in the Figure 2. We do not present it in the manuscript because this is full of information.*

9-The results and discussion section contains theoretical information on the used methods, which should be placed to appropriate section.

*We have moved the theoretical information to the methodology section (Page 9-12).*

10- Figure 4 requires more explanation. Requires labelling and a detailed figure caption.

*We remade this figure (page 27).*

11-In Table 1: The 2 mm/yr slip-rate for the Venta de Bravo fault could not be found in the related citation (Suter et al., 1995).

*We have changed the reference by Suter et al., 1992 (Page 29).*

12-Table 1 Add error ranges for slip-rates, fault length and scarps.

*We have added a few available slip-rates and their uncertainties along the PAFS (Page 29).*

13-Line 173-179 and 184-188: The purpose of these texts within the context of maximum magnitude is not clear.

*We have removed these lines and moved to 2.1Paleoseismology in the PAFS (173-179) and to 4.6Intensity Scale (184-188).*

[revised manuscript text omitted]

*Scarp refers to the top of the faulted rock unit.

Here, we reported the maximum earthquake magnitudes obtained with Wells and Coppersmith (1994). The ascending numbers are referred to in Figure 2.